# Design and Implementation of an HCPS-Based PCB Smart Factory System for Next-Generation Intelligent Manufacturing

Jinyoub Kim [1,2] , Dongjoon Seo [1] , Jisang Moon [1], Juhee Kim [2], Hayul Kim [2] and Jongpil Jeong [1,*]

[1] Department of Smart Factory Convergence, Sungkyunkwan University, 2066 Seobu-ro, Jangan-gu, Suwon 16419, Korea; kjy333k@naver.com (J.K.); srencat@hygino.co.kr (D.S.); yesmun1@hygino.co.kr (J.M.)

[2] Hygino AI Research Lab., 25, Simin-daero 248beon-gil, Dongan-gu, Anyang 14067, Korea; zan7702@hygino.co.kr (J.K.); hygino@hygino.co.kr (H.K.)

\* Correspondence: jpjeong@skku.edu; Tel.: +82-10-9700-6284 or +82-31-299-4267

**Abstract:** The next-generation intelligent smart factory system that is proposed in this paper could improve product quality and realize flexible, efficient, and sustainable product manufacturing by comprehensively improving production and management innovation via its digital network and intelligent methods that reflect the characteristics of its printed circuit board (PCB) manufacturing design and on-site implementation. Intelligent manufacturing systems are complex systems that are composed of humans, cyber systems, and physical systems and aim to achieve specific manufacturing goals at an optimized level. Advanced manufacturing technology and next-generation artificial intelligence (AI) are deeply integrated into next-generation intelligent manufacturing (NGIM). Currently, the majority of PCB manufacturers are firms that specialize in processing orders from leading semiconductor and related product manufacturers, such as Samsung Electronics, TSMC, Samsung Electro-Mechanics, and LG Electronics. These top companies have been responsible for all product innovation, intelligent services, and system integration, with PCB manufacturers primarily playing a role in intelligent production and system integration. In this study, the main implementation areas were divided into manufacturing execution system (MES) implementation (which could operate the system using system integration), data gathering, the Industrial Internet of Things (IIoT) for production line connection, AI and real-time monitoring, and system implementation that could visualize the collected data. Finally, the prospects of the design and on-site implementation of the next-generation intelligent smart factory system that detects and controls the occurrence of quality and facility abnormalities are presented, based on the implementation system.

**Keywords:** next-generation intelligent manufacturing; human–cyber–physical system; knowledge engineering; enabling technology; manufacturing domain technology; next-generation artificial intelligence; printed circuit board-based smart factory system





## 1. Introduction

Intelligent manufacturing is a general concept that has continuously evolved along with the development and integration of information technology and manufacturing technologies. In general, intelligent manufacturing involves digital manufacturing and digital network manufacturing and due to its recent rapid development and influential breakthroughs, intelligent manufacturing using the Internet, big data, and artificial intelligence (AI) [1–3] has also evolved [1–3]. Its fundamental goal is to increase competitiveness via ceaseless efforts to improve quality, increase efficiency, and reduce costs. Intelligent manufacturing systems are always human–cyber–physical systems (HCPSs), which are complex intelligent systems that are composed of human, cyber, and physical systems and aim to achieve a specific goal at an optimized level [4–6].

In other words, the essence of intelligent manufacturing is to design, configure, and apply HCPSs at various levels [7–14]. Currently, there is a trend of promoting the establishment of smart factories to increase competitiveness between small- and medium-sized

manufacturing industries worldwide. In addition, we are faced with high demands for improved product quality and efficient and quick market responses, which significantly increases the need for innovative upgrades for manufacturing sites. Currently, most printed circuit board (PCB) manufacturers specialize in processing orders from prominent companies, such as Samsung Electronics, TSMC, Samsung Electro-Mechanics, and LG Electronics. From the point of view of NGIM HCPS 2.0 systems, these prominent companies have been responsible for all product innovation, intelligent services, and system integration, with PCB manufacturers mainly playing a role in intelligent production and system integration.

Data management that is based on NGIM HCPS 2.0 systems is required, but there is a limit to the collection, analysis, and tracking management that can occur using only documentation and simple data management. Currently, quality inspection is conducted manually and takes a significant amount of time. Because the recording of production times and the analysis of processing procedures that occur during the manufacturing process is not happening, the traceability of manufactured products is impossible and a customer's ability to respond to quality issues is also reduced.

As a result, PCB manufacturers now efficiently collect and manage various data regarding manufacturing conditions, which are generated during the processing procedures and quality control using AI, integrated management and control (via MES implementation), and the automation of task assignment and real-time monitoring using data visualization, etc. In particular, it is critical to implement AI methods to automate quality inspection, predict and maintain equipment, and integrate management with MES. To implement a differentiated smart factory system that is efficient and suitable for PCB manufacturers, a system that is capable of collecting and accumulating standard manufacturing data must first be established. It is necessary to increase the efficiency of task assignment (which is currently inefficient) to remove the heavy workloads that are biased toward manufacturing managers. Furthermore, real-time data collection, accumulation, analysis, and utilization are required to support rapid decision-making, such as automatic task assignment. Second, it is necessary to boost productivity and product quality by implementing an AI method that is capable of analyzing and utilizing microscopic data for quality inspection. Third, work efficiency needs to be improved by applying MES functions that are optimized for individual companies to create web-based smart factory systems that integrate manufacturing management, quality management, facility management, and data visualization to improve data accessibility and operational efficiency by monitoring the progress of each stage of production, from order to shipment. Fourth, real-time data visualization is needed to efficiently manage entire manufacturing sites by monitoring and managing work status, facility management, quality inspection data, and abnormalities in real time. In this paper, we propose an NGIM HCPS 2.0-based smart factory system comprising an alarm level-based double verification framework that is applicable to heterogeneous equipment, a deep learning-based algorithm, a web-based integrated MES, and data visualization for real-time field management. This paper makes the following specific contributions:

1. The proposal of a cloud–fog–edge distributed network that can connect heterogeneous devices in industrial sites using fog- and edge-based (rather than cloud-based) central control in order to address the bottleneck;
2. The proposal of a knowledge distillation-based algorithm that can efficiently apply deep learning-based algorithms that require a large amount of computing resources;
3. The proposal of an NGIM HCPS 2.0-based MES function that is optimized for PCB manufacturers to help them grow into world-class small- and medium-sized businesses by increasing global corporate competitiveness in the twenty-first century.

This paper is organized as follows. The technical framework and core technology are reviewed in Section 2. The core technology of the smart factory system that was designed for PCB manufacturers is detailed in Section 3. Section 4 describes the smart factory system that was implemented and Section 5 presents our conclusion and future challenges.

## 2. Related Work

### 2.1. PCB Manufacturing Status and Characteristics

PCBs are made up of conductors (usually copper) that can transmit electrical signals and insulating layers (usually phenol/epoxy [15]). The conductors provide electrical connections and holes in the layers allow for interconnections between the layers.

The manufacturing process of PCBs is divided into a complex configuration, from computer aided design (CAD) and computer aided manufacture (CAM) to the inspection process. Figure 1 depicts the general PCB manufacturing process [16].

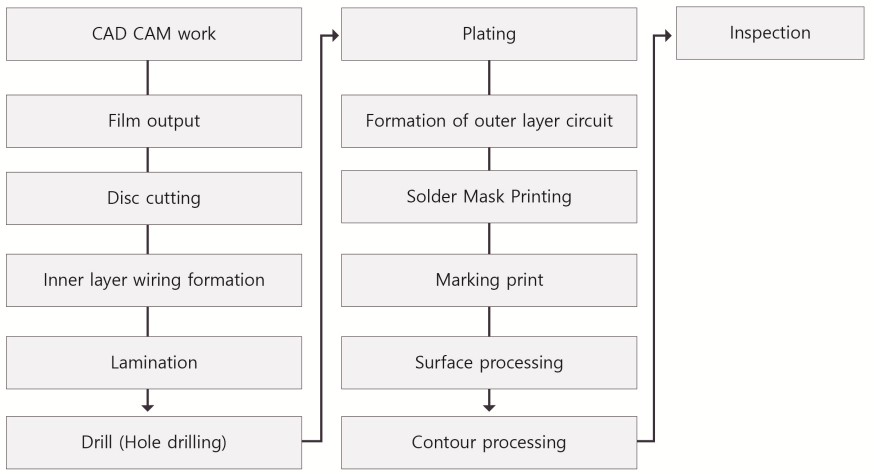

**Figure 1.** The general PCB manufacturing process.

### 2.2. NGIM HCPS 2.0 Systems

NGIM HCPS 2.0 systems have not only introduced revolutionary changes in creating, accumulating, utilizing, imparting, and inheriting manufacturing knowledge, but they have also significantly improved the ability of manufacturing systems to deal with uncertain and complex problems, which has resulted in significant changes in manufacturing system modeling and decision-making that could lead to progress [4]. The configuration of NGIM HCPS 2.0 systems is shown in Figure 2.

For example, when using an NGIM HCPS 2.0 system for an intelligent machine tool, a digital model of the entire machining system can be built through a process of sensing, learning, and recognition, which results in high machining quality and efficiency, as well as low energy consumption [17,18] and high-precision machining. This process can be used to optimize and control the entire machining procedure. NGIM HCPS 2.0 systems should be used to comprehensively upgrade all manufacturing activities, including research and development (R&D), production, sales, services, management, and system integration, in order to substantially increase quality, efficiency, and competitiveness. In other words, the essence of NGIM is to develop and implement various HCPS 2.0 systems for various purposes to provide innovative improvements in social productivity and integrate those improvements into a network of HCPS 2.0 systems. HCPS 2.0 systems can be regarded as universal solutions that are capable of effectively addressing the challenges that are associated with manufacturing industry innovation and upgrades as they can be widely applied to product innovation, production innovation, and service innovation within manufacturing and process-oriented manufacturing. The development of HCPS 2.0 systems is expected to proceed as follows: the use of HCPS 2.0 systems to enable manufacturing systems with next-generation AI technologies. There are several approaches to developing innovation-driven manufacturing engineering, but two stand out in particular. The first is an original innovation for fundamental and, crucially, manufacturing technology. The second approach is the use of common supporting technologies to promote manufacturing technologies, which can lead to the development of innovative manufacturing technologies

by combining the two types of technologies and can also be used to upgrade various manufacturing systems. This type of innovation is transformative, integrative, and universal. The common supporting technologies of the last three industrial revolutions were steam engines, electric motor technology, and digital technology. AI technology has become a common supporting technology in the fourth industrial revolution [1]. The integration of these generic technologies with manufacturing technologies can promote innovation and upgrades within manufacturing. As a result, NGIM that is based on HCPS 2.0 systems could be a significant driver of innovation-driven development in the manufacturing sector, as well as a significant roadmap for innovation and upgrades. However, next-generation AI technologies must be thoroughly integrated with manufacturing technologies to create NGIM technologies. Because manufacturing technologies are the foundation for upgrading manufacturing processes and are used to enable technology, supporting technologies can only provide full scope when they are integrated with manufacturing technologies. In summary, manufacturing technologies are the basic foundations and supporting technologies are the devices and systems that make manufacturing possible. As a result, the dialectal unity and integrated development of these skills are required. From an intelligent technology standpoint, NGIM can be viewed as a way to promote and apply advanced information technology. However, from the standpoint of manufacturing technologies, NGIM can be viewed as a way to promote innovation through the use of supporting technologies that enable generic technologies and encourage the upgrade of manufacturing systems in various industries.

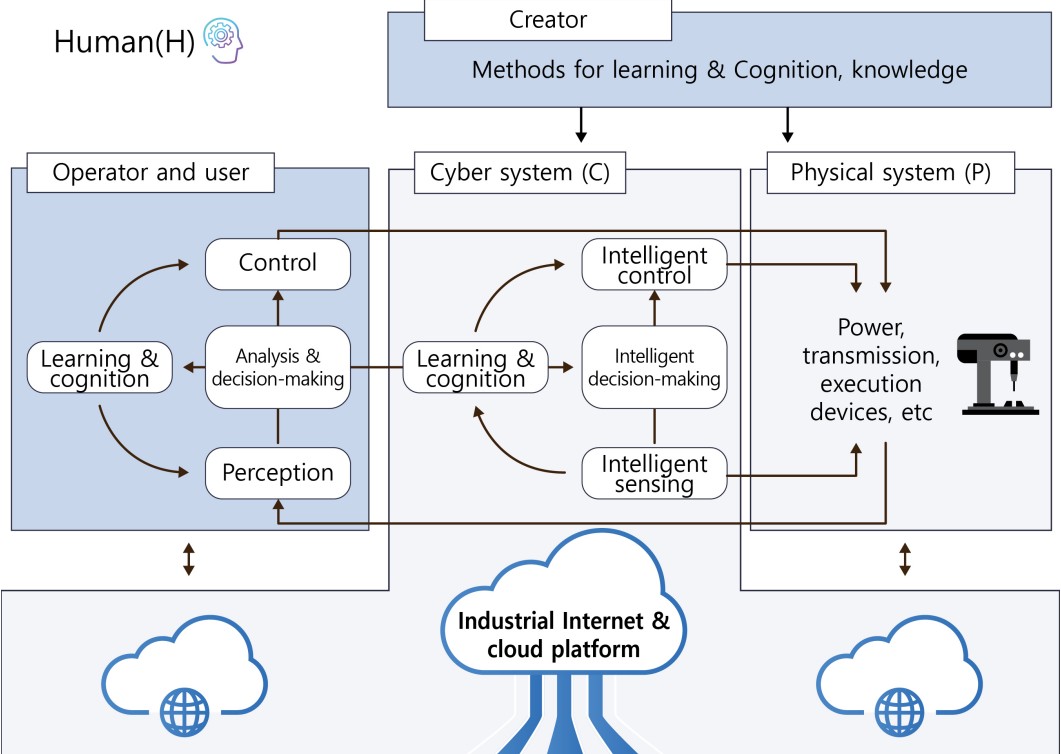

**Figure 2.** The configuration of NGIM HCPS 2.0 systems.

Manufacturing domain technology refers to technologies that are related to the physical systems within HCPSs. These include general manufacturing skills and specialized domain skills [19]. Intelligent manufacturing originated from the manufacturing industry. As a result, manufacturing technologies are the foundation of HCPSs for intelligent manufacturing. Intelligent manufacturing not only encompasses manufacturing and process-oriented manufacturing but also product life cycles. Therefore, it covers a wide range of manufacturing domain technologies [20] that can be classified according to their perspec-

tives. For example, from the standpoint of manufacturing processes, these technologies include plastic forming technologies, cutting technologies, casting technologies, welding technologies, heat treatment technologies, and additive technologies, among others [21–25]. Machine intelligence technology is related to HCPS 2.0 cyber systems, which are based on the integration of AI technologies with manufacturing domain knowledge to achieve specific HCPS goals. The cyber systems guide the HCPSs by assisting humans with the necessary awareness, analysis, decision-making, and control for the HCPSs so that the physical systems can perform optimally. Intelligent sensing, autonomous recognition, intelligent decision-making, and intelligent control are the four major categories of machine intelligence technology.

Intelligent sensing is a basis and prerequisite of cognitive learning, decision-making, and control. Its goal is to effectively acquire all types of internal and external information, including data collection, transmission, and processing. Detection design, high-performance sensors, and real-time intelligent data collection are all important technologies [26,27]. Autonomous awareness ensures that the systems acquire the knowledge that they need to achieve their objectives. This task is critical for effective decision-making and management. Cognitive tasks in HCPS 2.0 systems are generally completed by the collaboration between cyber systems and humans. As a result, it is necessary to address any issues that concern intelligent machine autonomy and human–machine collaboration. System modeling is another important task in the autonomous recognition process of intelligent machines (including parameter identification). The core skills include the self-learning of model structures and model parameters, model evaluation, and self-learning optimization [23]. Intelligent decision-making evaluates the state of the systems and determines their optimal operation. Decision-making tasks in HCPS 2.0 systems are generally completed by the collaboration between cyber systems and humans. As a result, it is necessary to solve any problems that are associated with intelligent machine decision-making and human–machine collaboration. Key intelligent decision-making techniques include accurate system assessment, decision-making model optimization, and predictive decision risk analysis [28].

Intelligent control adjusts the systems according to their decisions to achieve their goals. This task is required to solve problems with human–machine cooperation and machine autonomy. Dealing with the uncertainty of the systems and their environments is a key challenge in intelligent control, along with developing intelligent control technologies, such as adaptive control [12,13]. Human–machine collaboration technologies raise many uncertain and complex issues that intelligent manufacturing cannot solve solely using human or machine intelligence. A typical feature of next-generation AI is human–machine hybrid augmented intelligence. The core critical technology of NGIM (i.e., HCPS 2.0) includes human–computer interaction technologies [11,24–28], as well as cognitive, decision-making, and control-level human–machine collaboration.

### 2.3. Cloud Manufacturing Technology

Cloud computing is a new service-oriented computing technology that has emerged in recent years [14,29]. In cloud computing, highly virtualized computing resources are organized using a cloud computing platform and a large-scale resource pool is formed to provide unified services. Individuals and enterprises can access computing resources on-demand through heterogeneous and self-governing Internet services. It is possible to respond quickly to changes in demand and work remotely within the same system environment, regardless of time or location. Figure 3 shows the structure of a cloud system.

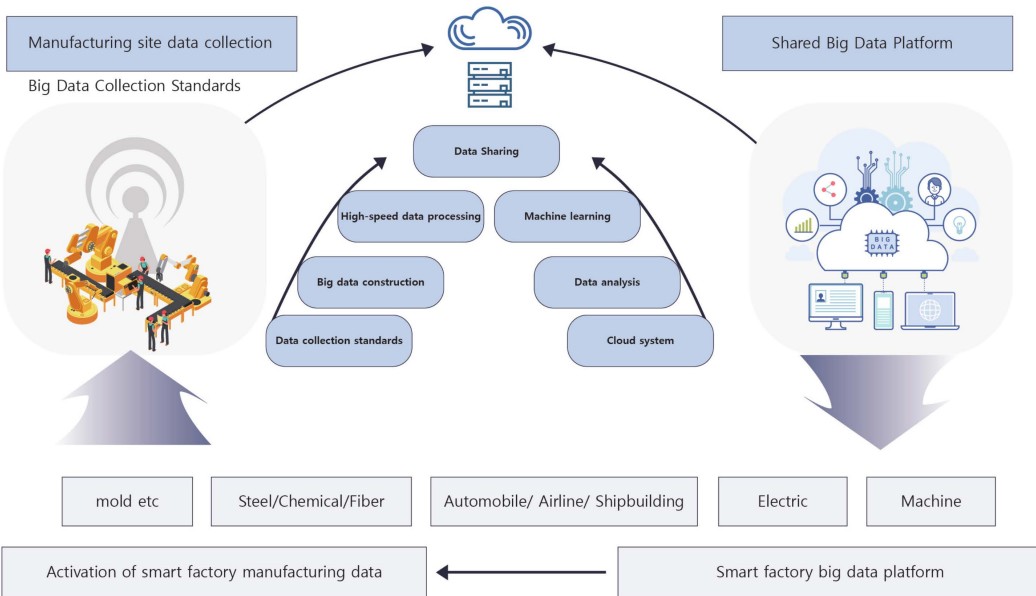

**Figure 3.** The structure of a cloud manufacturing system.

A cloud manufacturing system consists of manufacturing resources and capabilities. Those manufacturing resources and capabilities are virtualized and oriented toward service provision. In cloud manufacturing, the pervasive and efficient sharing and coordination of resources and capabilities can be achieved using unified and centralized intelligent management and operation. Cloud manufacturing system also analyze and divide service requests and automatically search the cloud for the best-matched services. Using a series of processes (including scheduling, optimization, and combination), a solution is generated and sent back to the client. The user does not need to communicate directly with every service node, nor find the specific locations and situation of the service nodes. Through the cloud manufacturing platform, manufacturing resources and capabilities can be used in the same way as water, gas, electricity, etc. [30].

### 2.4. Data Visualization

The visualization layer handles a variety of tasks, such as operational dashboards, control and governance, data analytics, portal and mobile usage, and application programming interface (API) gateways. Operational dashboards display all of the relevant sensors, devices, and machinery in real time. They also include the basic standard operating functions of the manufacturing units. Control and governance are controlled from the command center, based on the real-time monitoring or anomaly detection of machines and sensors [31]. Data analysis is the process of examining data to identify patterns or trends. Data analysis and machine learning algorithms are also used to perform predictive and preventative actions. Analytics and metrics for portal and mobile usage are presented on graphical user interfaces on consumer applications or mobile devices. The API gateways display enterprise applications for demand forecasting, inventory management, traceability, and other purposes and they also participate in the orchestration of business process management (BPM).

## 3. HCPS-Based Smart Factory Model

### 3.1. System Architecture

Intelligent manufacturing systems are complex intelligent systems that are made up of humans, cyber systems, and physical systems, which work together to achieve specific manufacturing goals with high levels of efficiency. This type of intelligent system is known as an HCPS. HCPSs can be used as technological principles and architecture designs for intelligent manufacturing. It is possible to conclude that the essence of intelligent manufac-

turing is to design, configure, and apply HCPSs in a variety of situations and at various levels [28,32–34]. With the development of information technology, intelligent manufacturing has progressed through the stages of digital manufacturing and digital network manufacturing and has evolved into NGIM. NGIM incorporates advanced manufacturing technologies (i.e., root technologies) with next-generation AI technologies (i.e., enabling technologies). It is the primary driver of the new industrial revolution. Figure 4 depicts the structure of our PCB smart factory system, which was based on an NGIM HCPS system.

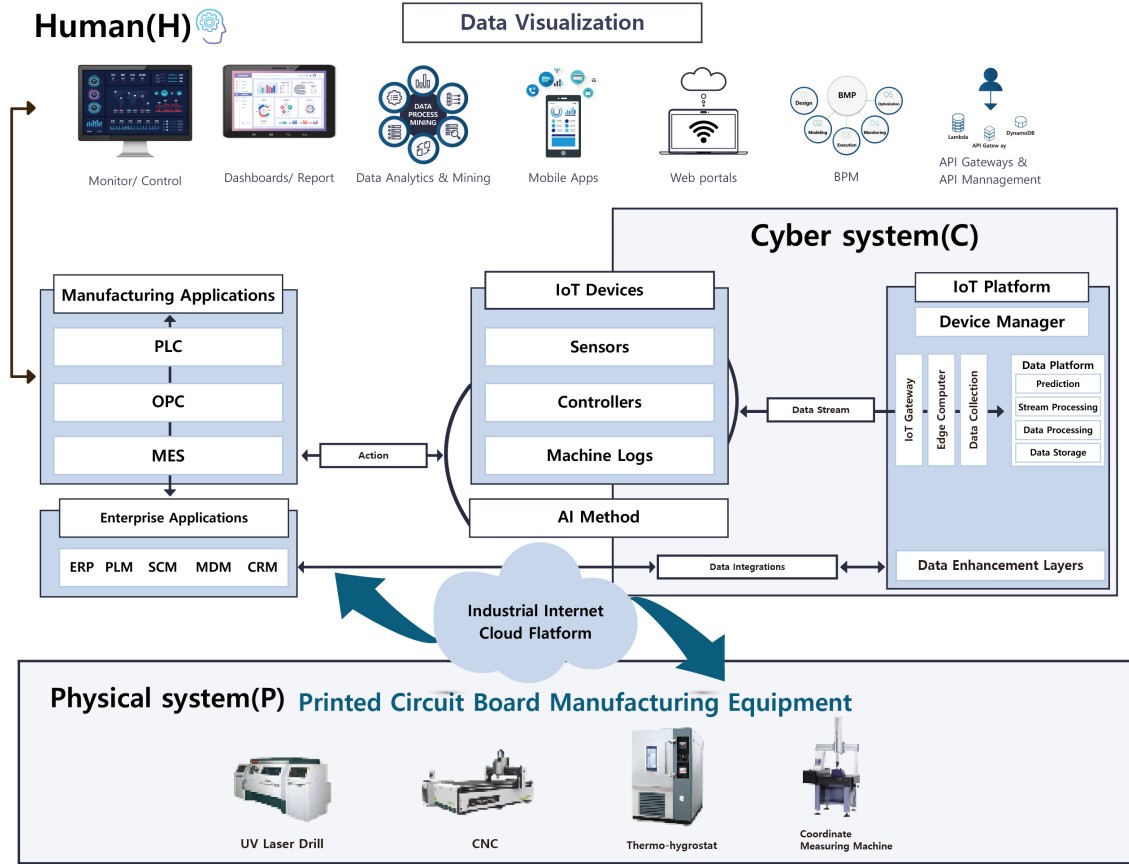

**Figure 4.** The architecture of our HCPS-based PCB smart factory system.

In this paper, the raw data were from a more advanced system, which is currently required at the manufacturing sites of most PCB manufacturing companies following the fourth industrial revolution. We discuss the core technology that was designed and implemented from the standpoint of HCPSs and the intelligent manufacturing system that was implemented to make it digitized.

### 3.2. Data Gathering

Data gathering refers to the collection and management of data that were generated by Internet of Things (IoT) devices and AI methods at an advanced level through an IoT platform. IoT platforms consist of several layers and components. These platforms are critical for transforming factories into smart factories. At a high level, IoT platforms must have the ability to extract data from equipment, sensors, devices, and AI methods, examine large amounts of data using edge analytics in real-time, and grow with minimal costs. They also need to store large amounts of data.

Secure industrial demilitarized zone networks (IDMZs) [35] for manufacturing and enterprise applications, in which sensors and devices are typically connected in factories, house IoT gateways. The devices are IP-capable and can be uniquely identified on the

networks. As the costs of sensors and wireless protocols rapidly decrease, sensors become more affordable and ubiquitous, thereby allowing wireless mesh networks [36] to flourish. The establishment of continuous connections between the sensors and equipment is critical to these networks, so the sensors in factories are typically wired using Ethernet cables (TCP/IP) or connected wirelessly (RFID/ZigBee/Bluetooth).

Edge computing consists of gateway servers or router services that perform the real-time computing that is necessary to make fast local decisions about data streams for low-latency control. Edge computing configures machines so that they are closer to devices for low-latency operations and does not wait for decisions from the subsequent layers of the data lakes. The decision services also work with the device managers to send control parameters to programmable logic controllers (PLCs) or open platform communications (OPCs) for system control and optimization.

Data ingestion refers to a data ingestion layer in which data are processed and further transformed either by being streamed in real time or in batches from multiple source applications. Data from multiple sources in various formats (e.g., time series, event streams, log streams, structured, semi-structured, unstructured, etc.) can be converted into the standard format for individual enterprises. The data serialization format (for example, Protobuf, Avro or thrift) is chosen for its speed and consistency. Data quality and harmonization must be considered, depending on how the data are maintained in the original application. To receive massive amounts of data, reusable data pipelines can be installed using either an Apache Kafka cluster or a waterway.

Data lakes are stored in Hadoop distributed file system (HDFS) clusters, relational database management systems (RDBMSs) (e.g., Oracle, MySQL, MS SQL, etc.), and NoSQL databases (e.g., Cassandra, Mongo, etc.), depending on the data and usage type. Apache Spark is used for real-time analytics and is several times faster than MapReduce. For structured data processing, data are stored in relational format using Spark SQL. Spark scripts that are written in Scala, Python or Java process and analyze semi-structured or unstructured data. Spark includes a fundamental machine learning (ML) library that trains and tests datasets, creates reusable pipelines, and applies prediction or clustering algorithms.

Data integration, enterprise applications, and manufacturing applications are typically connected using middleware and extract, transform, load (ETL) tools. Data from these various enterprise systems are extracted into a data lake or data ingestion layer using middleware or ETL tools. Data are then typically processed using a series of steps that are called data preparation areas, in which the data are enhanced, transformed, and enriched into a standardized and sharable form. Before being fed into the lake, the data are typically combined with multiple IT applications within the realm of data staging. There are several commercial and open-source ETL products that are available for data integration, which can migrate and transform massive amounts of data. When the information technology (IT) application environment is primarily cloud- or SaaS-based, commercial integrated platform as a service (iPaaS) products are more appropriate.

*3.3. AI Method*

Currently, the majority of PCB manufacturing companies that perform hole processing using $CO_2$ laser equipment on packages, F-PCB, and non-memory semiconductor substrates capture images of hole units using high-magnification microscopes and hole size measurement programs and then manually (visually) inspect and measure the states and sizes of the holes. The upper and lower circles that are produced by PCB hole processing range in size from 40 μm to 100 μm (about 0.04 mm to 0.1 mm). One PCB panel can contain 700 million to 8 million holes, which were made during the hole processing. However, human visual inspection only covers a small portion of the holes and it is anticipated that hole sizes will continue to shrink in the future, which would limit the success of human visual inspection. The AI method that we applied to implement our NGIM HCPS-based smart factory system could be managed in real time to overcome these limitations and improve productivity due to reduced inspection times and facility utilization rates, the systematic

image storage management and measurement information, and its integration with an MES system. This study considered that preprocessing to remove noise was essential for computers to recognize circle shapes accurately because the human visual inspection of images that were collected with high-magnification microscopes can identify circle shapes but computers cannot. Therefore, we intended to maximize the efficiency of our system by using an algorithm to detect circles and a deep learning-based algorithm to analyze and learn the detected images.

Current algorithms for detecting circles include the Hough circle transform (HCT), the histogram of gradient and texture segmentation using the Gabor filter, and random sample consensus (RANSAC). Of these, RANSAC was able to accurately detect circles even in noisy parts of the original images.

A semantic segmentation model was introduced to train this process. Segmentation methods that are based on upsampling or deconvolution mainly include FCN, U-Net, and SegNet. U-Net is a semantic segmentation network that is based on FCN, which can be trained using very few images and can outperform the previous best networks using the ISBI database of cell image segmentation by electron microscopy [37]. The U-Net network was designed for biomedical images and has been widely used in medical image segmentation since it was first proposed [38].

Our primary focus when selecting an algorithm to use was addressing "speed, context, and localization" issues. The U-Net structure has no fully connected layers (only convolutional and downsampling layers) and the network can simultaneously combine low-level and high-level information (Liu et al., 2021). Due to this feature, although the current segmentation models create a type of computational waste that slows down the process, U-Net increases the speed because of the low overlap ratio of the unit that identifies the images. There are currently few cases in which this AI solution has been implemented in PCB hole processing. Therefore, by using U-Net (which is a deep learning network that is renowned for its efficiency in semantic segmentation within the field of medical artificial intelligence), this study became the first case of integrated medical AI and PCB manufacturing engineering. The U-Net approach was employed in this study and the accuracy increased to 96.7% (when utilizing the AI method) from 87% (when utilizing on-site human visual inspection). This meant that U-Net adequately fulfilled its function in our NGIM HCPS 2.0 system, so we used it as the algorithm for the "AI technique", among other methods that utilized different residual 3D CNN networks.

Our future study will compare these different methods, including PSP-NET, RES-NET, etc., with the aim of achieving 99% accuracy. An algorithm from the classification field could be used to recognize the different types of defects on PCB substrates. Instead of concentrating on the AI method, this study concentrated on demonstrating our new NGIM HCPS 2.0-based smart factory system that could be used by PCB manufacturers by successfully integrating the modules of each layer, as depicted in Figure 4. The development of new algorithms and the advancement of U-net utilization via parameter adjustments are two areas of additional research for the AI approach that could be explored in the future.

The structure of our data gathering processor is depicted in Figure 5. Following construction, the retrieved original images were stored according to LOT and date. The preprocessed images were kept once the preprocessed images and the AI measurement images were combined. After creating a measurement directory for each date, the measurement images were saved and the analysis information (such as positive/non-determination information and size information) was updated in the database by analyzing the LOT reference information (MES integration) and measurement sizes. Separate AI preprocessing, hole size measurement and learning, and analysis data were needed in addition to the original images that were captured by the microscope. The findings from our image information analysis using the AI method were linked with reference data for each LOT using the MES and the results were recorded in the database once the data consistency was verified.

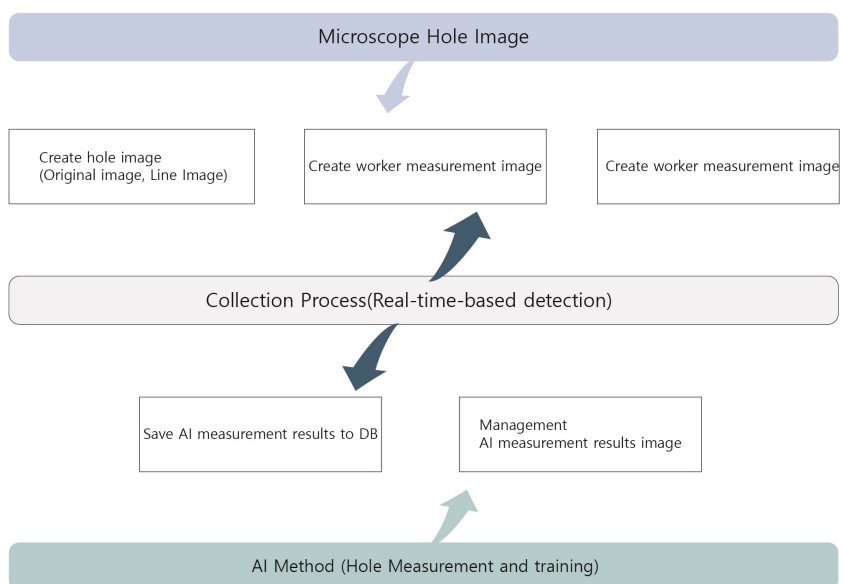

**Figure 5.** The architecture of our data gathering processor.

Figure 6 depicts the structure of our image inspection system, which employed the AI method to transmit data from the microscope in the field to the data collection server to generate the image files and transcripts and collect the data. It continuously monitored the image files and transcripts and transmitted them to the AI algorithm's file transfer protocol (FTP) server while also storing the data in the database. Holes in the image files for which the hole sizes were not measured using data in the database were measured using the AI algorithm. The images were then divided into "preprocessed images" and "AI measurement images" and stored on the FTP server. The algorithm provided the analysis data on the monitoring screen by combining data from the image files and the database.

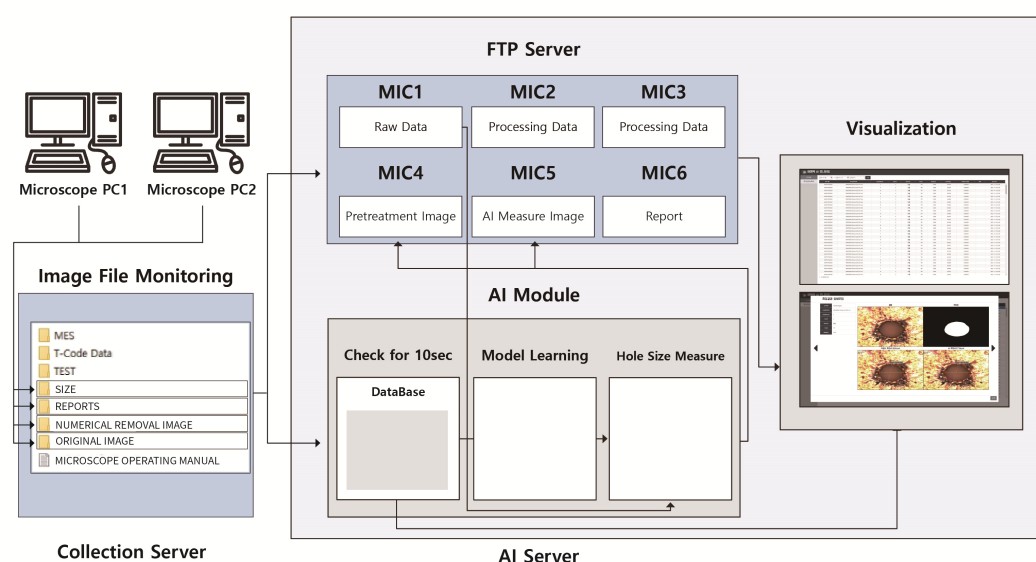

**Figure 6.** The structure of our inspection system when using the AI method.

### 3.4. Manufacturing Applications

Manufacturing execution systems (MESs) are online transaction processing (OLTP) systems that record all transactions in the field, such as material movement, input, rework, scrap, etc. They also provide information. Manufacturing applications can create their own

in-house MESs to meet the needs of the individual companies. They offer ready-to-use systems for critical functions, such as LOT traceability and LOT lineage, serial number traceability, test data capture, label printing, etc. However, when specific companies have very specific requirements, they must weigh up the distinct advantages and disadvantages of the two methods to determine which would be best for them [39]. An overall business process flow diagram is depicted in Figure 7.

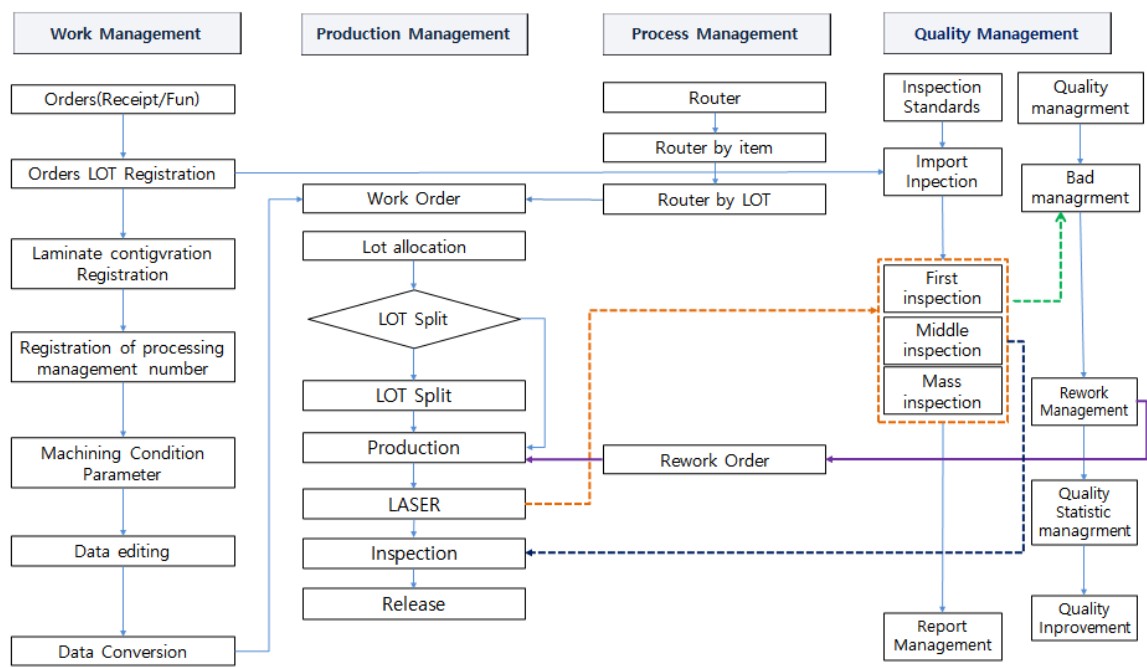

**Figure 7.** A flow diagram of an overall business process.

Programmed logic controllers (PLCs) manage the coordination between equipment, process steps, and operators to produce finished products. There is usually more than one PLC in pieces of production equipment. The equipment suppliers program the PLCs when they supply the equipment [40–43]. Open platform communication (OPC) is a middle layer that allows for communication between MESs and PLCs. As previously stated, MESs record all transactions that are performed in the field while PLCs control the equipment that is used to carry out the process steps. To record the transactions in real time, MESs must constantly communicate with the PLCs and work closely with the equipment and other systems to act as "supervisors" in the field. They are also used to control devices remotely [44].

*3.5. Enterprise Applications*

Enterprise applications consist of the various IT applications that are used to support and run businesses [45]. They are primarily made up of applications for PLM, enterprise resource planning (ERP), and supply chain management (SCM), as well as other custom applications. ERP provides an integrated platform for running multiple business processes, such as manufacturing, purchase to pay, order to cash, planning, accounting, costing, integration, inter-company transfers, etc. Product life cycle management (PLM) is the process of consistently manages the entire life cycle of a product, from product design to the production of the final product, in order to increase the added value of the product and reduce costs. product life cycle management (PLM) systems provide product data, management server systems, and network systems for multiple client systems. SCM is a

packaged application that focuses on supply chain planning, predictive management, and production planning.

### 3.6. Data Visualization

We developed a system that could collect and monitor production results in real time by connecting ERP systems, MESs, sensors that are attached to equipment, PLCs, and machine log files using data visualization. Figure 8 shows a structural diagram of the smart factory monitoring system.

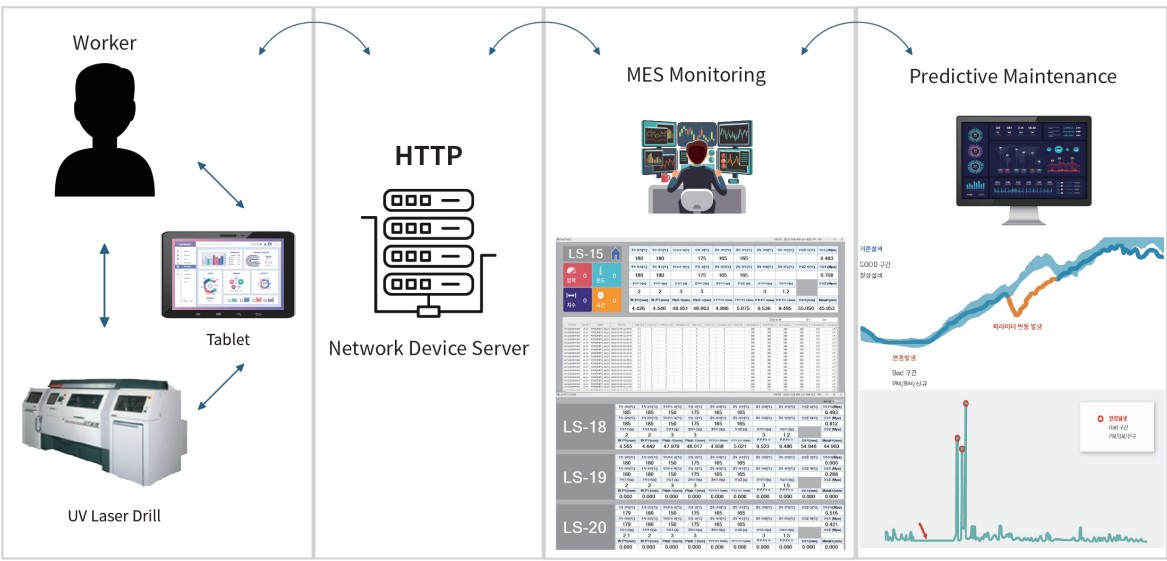

**Figure 8.** The structure of our smart factory monitoring system.

Monitoring systems in smart factories connect equipment and machine log files to transmit the collected information to database servers via intermediary collection devices, such as tablets. Before beginning work, operators select work orders using tablets and then start production. The production performance data that are collected from the production facility are matched with the work orders and are then transmitted to the database servers through the tablets. The production performance data are stored in ERP systems in conjunction with in-house MESs. Managers can monitor the real-time production status through the monitoring systems and when abnormalities occur, the managers can see that information in real time.

## 4. Implementation and Results

### 4.1. Implementation Environment

The structure of our PCB smart factory system was defined and Figure 9 depicts the configuration of the hardware (H/W) system that was and the H/W system architecture. The configuration was broadly classified as follows: server operations; office or field operations; manufacturing application operations for workers; IoT operations; and real-time feedback operations.

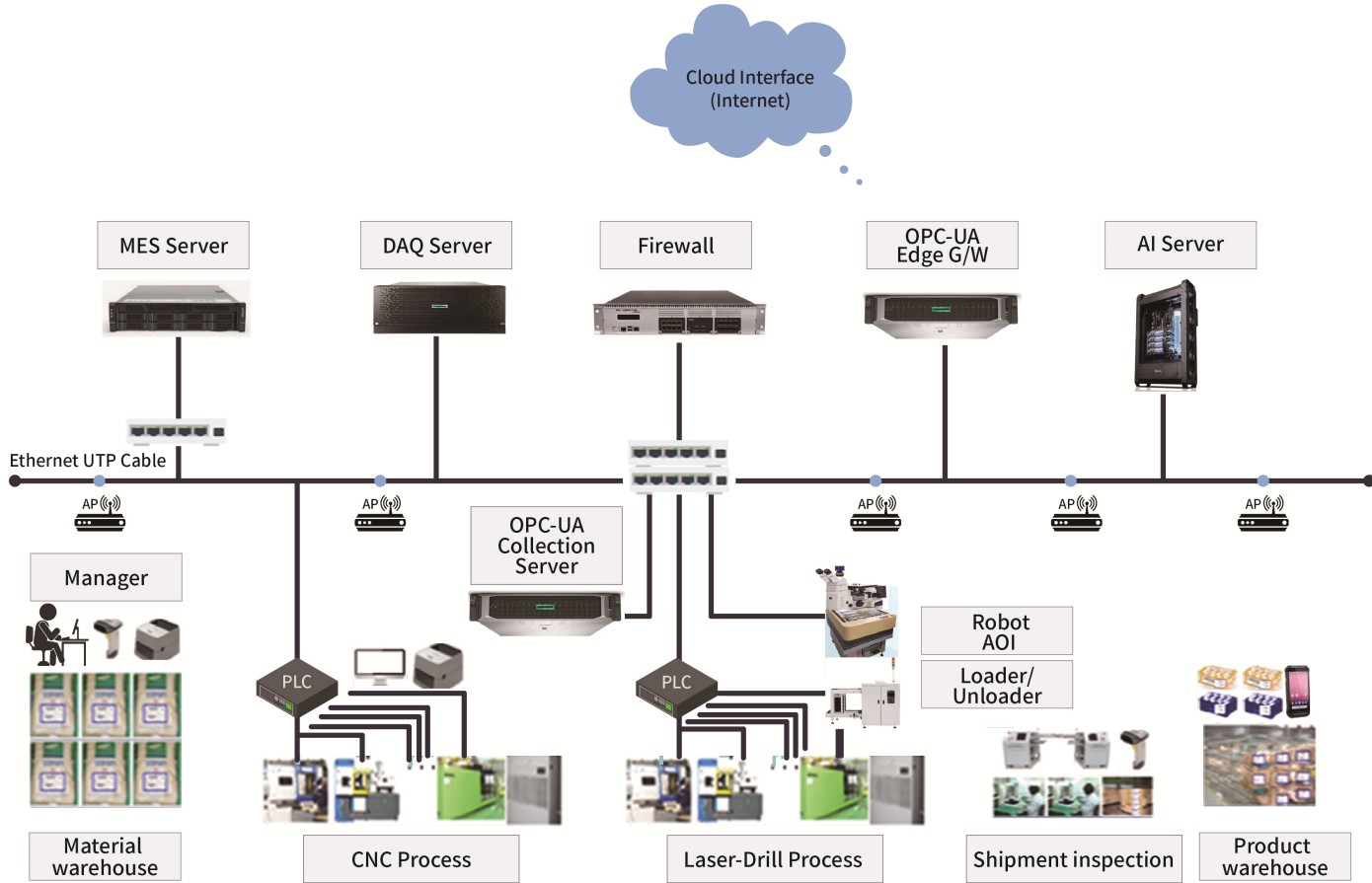

**Figure 9.** The hardware configuration.

The hardware was set up to build and run an MES on a cloud platform that was simple to secure, use, and manage. Keyed in data were automatically collected and sent to the MES or data acquisition (DAQ) server via a monitor or a manufacturing application that was run using this information. In addition, it was configured to enable real-time monitoring and to manage abnormal occurrences and support real-time decision-making. The smart factory software was created by integrating all manufacturing-related processes, from product order to shipment. It integrated all aspects of factory operations, not only including application systems but also field automation and control automation. Figure 10 depicts the software (S/W) configuration with these characteristics.

Various IoT data from different devices, such as PLCs, barcode readers, temperature and pressure sensors, machine logs, etc. were automatically collected using OPC-UA, FTP, and TCP/IP by the edge computer. The smart factory software is created by integrating all manufacturing-related processes, from product order to shipment. It will integrate all aspects of factory operations, including not only application systems but also field automation and control automation [42,43,46–48]. Figure 11 depicts the S/W configuration diagram with these characteristics in mind.

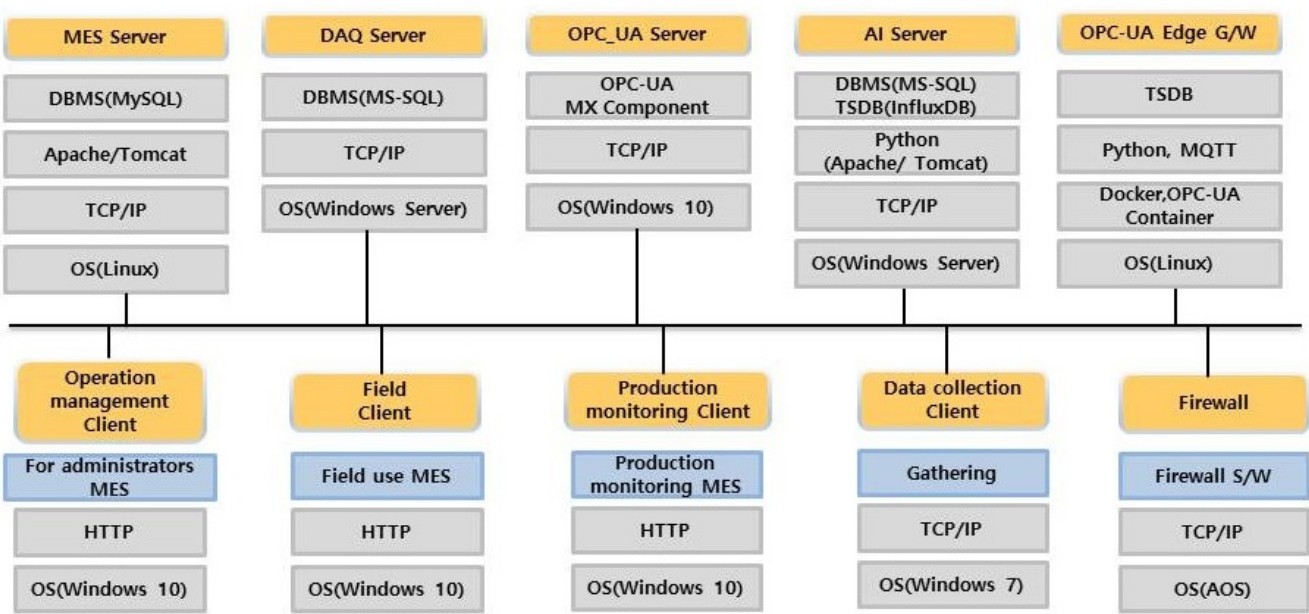

**Figure 10.** The software configuration.

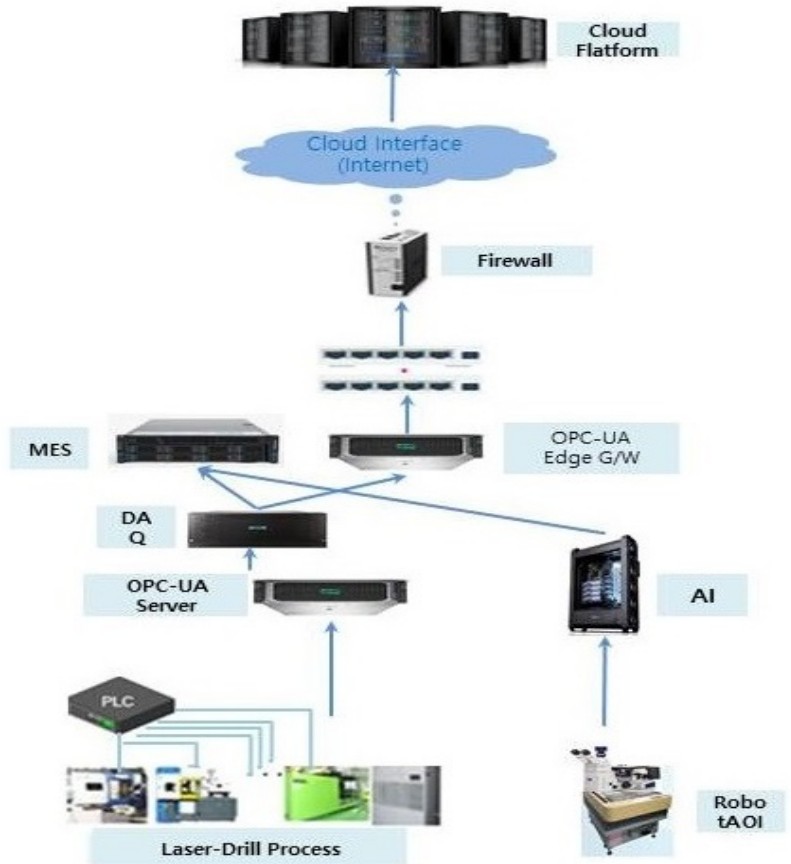

**Figure 11.** The configuration of the data gathering system.

*4.2. Evaluation Metrics*

By implementing our integrated manufacturing application that was based on HCPS 2.0, improvements in the productivity, quality, sales, and profit growth of a company were realized using the evaluation indicators to measure whether each goal was achieved. Target

values were presented in the first half of 2021 and the improved indicators were set and managed for each year or specific period. Table 1 shows the KPIs that were presented in the first half of 2021 and the facility utilization rate represents the increase/decrease in the number of operating hours for production equipment, based on there being 24 h in a day. Excluding downtime due to decreases in order volume, the operation rate was calculated as (actual facility operating time/worksite working hours) × 100. The actual operating time refers to the time that the equipment spent emitting laser beams and the working hours are based on there being 24 h in a day. When unprocessed downtime occurred due to insufficient order volumes, the unprocessed downtime was calculated separately and managed as a key performance indicator for the sales department, which was the time that was registered on the production calendar but excluded downtime due to unavoidable external events or when there were no products to be processed. The process defect rate represents the number of defects in relation to the production input per process unit and was calculated as the number of generated defects/process input. The unit of production is the production scale meter and the unit of defective quantity is the defective square meter.

**Table 1.** The KPIs of facility utilization rate and process defect rate.

| Field | Key Performance Indicator (KPI) | Unit | Current | Target | Weight | Remarks |
|-------|---------------------------------|------|---------|--------|--------|---------|
| P | Facility Utilization Rate | % | 70 | 82 | 0.7 | Laser drilling process, based on 24 h in a day |
| Q | Process Defect Rate | PPM | 1100 | 900 | 0.3 | |

To store the facility's performance data in the form of machine log files in real time, the IoT platform communicated with the PC that was installed in the facility in real time and successfully transmitted the data to the designated manufacturing management data server (DAQ, MES or edge computer) using IP communication. To analyze the outcomes, real-time monitoring was carried out using data visualization, which consisted of inspection progress by unit, LOT status by process, progress status by LOT, and facility operation/non-operation status items. The log file status by facility, log transmission status by facility, and facility data management were all aspects of IoT data management. Each abnormal occurrence and quality inspection data point used the prediction/maintenance status and a real-time decision support system was designed and developed to implement a production site prediction/maintenance system by linking the task assignment status with the work order by facility item, as well as abnormal occurrence management. We then checked that it worked properly.

*4.3. Implementation Results*

The MES was largely divided into the management, field, production monitoring, mobile, OPC, and AI methods, according to function. Figure 12 shows the MES when divided by function. All processes, such as product production, equipment, quality, manufacturing conditions, outsourcing, and management, were covered by the MES for managers. The on-site MES used an MES in the field to perform certain functions, such as performance processing and conditions management, and monitoring is carried out so that the collected information, such as production status and processing status, could be checked in real time. The mobile MES improved user access to the MES by using a personal digital assistant (PDA). It used OPC to collect equipment data and then ran a series of processes to control the equipment. The AI method performed a non-destructive quality inspection function by measuring the PCB hole sizes.

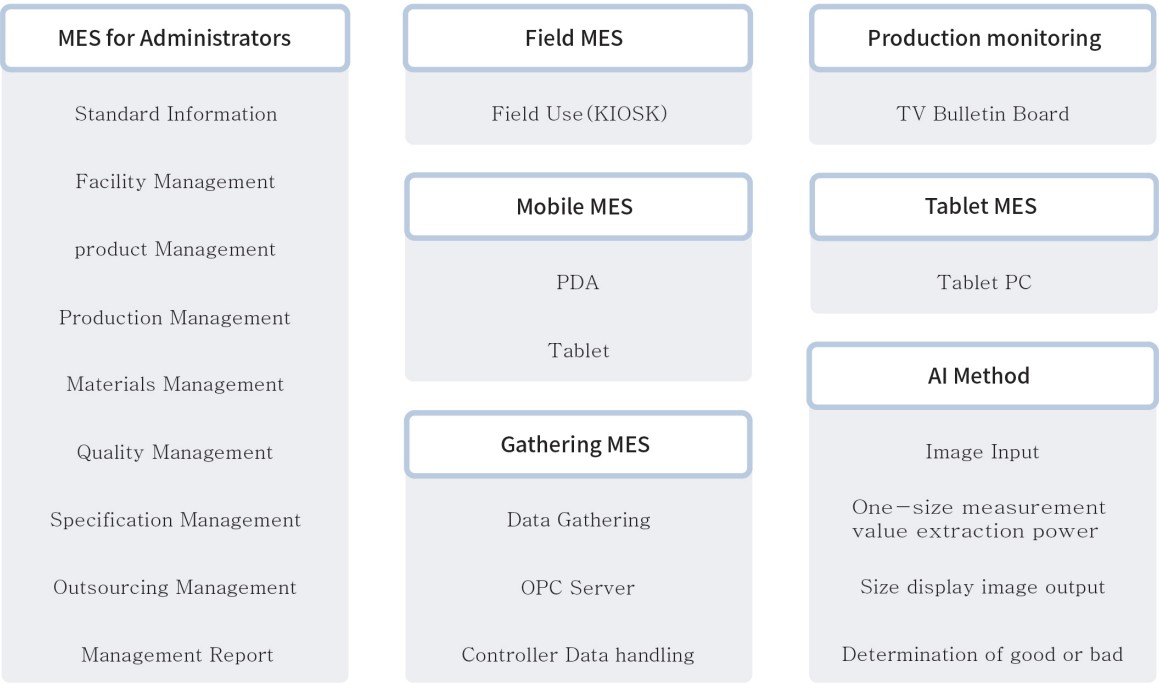

**Figure 12.** A functional diagram of the MES.

A flow chart for each major process is shown in Figure 13. The order information was received via phone, messenger, and text and order confirmation entailed photographing and registering customers they entered the customer warehouse.

LOT warehousing referred to LOT warehousing registration that used the PDA to register the customer management number and the LOT number using an optical character reader (OCR) function after the LOT check operation, in which the manager could visually confirm whether the customer registration, product, LOT number, and customer management number match was complete. For a product to be issued, the company management number needed to be confirmed, the preprepared items needed to be checked, and the company management number needed to be generated. Import inspection was classified into two types: the visual inspection of defective items to register defect quantity; the measurement inspection of items to register, compare, and confirm product. There were also two types of LOT assignment: C-side assignment, which performed unit assignment after checking the preliminary items but before registering the C-side assignment unit; S-side assignment unit, which performed unit assignment after checking the preliminary items but before registering the S-side assigned unit. When there were no problems after comparing the product, size, and customer size, the C-side preparation for product loading, machining conditions, machining data confirmation, and machining registration was initiated using the PDA. For the C-side process inspection, there were three types of inspection: the first inspection, heavy inspection, and final inspection. Microscopic inspection (visual inspection), surface size measurement inspection, floor size measurement inspection, defect input, and inspections of the PNL display were all performed in each case.

After the initial inspection, heavy inspection, and final inspection, the quantity of PNL that was being sent out after inspection was checked and, after re-inspection, the completion of C-side processing and the inspection confirmation of the unloaded C-side products were registered using the PDA. When there were no problems after comparing the product, size, and customer size, the S-side preparation for product loading, machining conditions, machining data confirmation, and machining registration was initiated using the PDA. For the S-side process inspection, there were three types of inspection: first inspection, heavy inspection, and final inspection. After the initial inspection, heavy inspection, and final inspection, there were also S-side inspection confirmations to check the quantity of PNL and the PDA was used to register the S-side processing completion and to unload the S-side products. Shipment inspection included C-side, S-side, and PNL re-inspection. The PNL re-inspection continued with the processing status confirmation and inspection of the PNL quantity, but the process inspection confirmation was defective. Work on re-confirming the PNL and measurement data is currently underway. Following the LOT registration, the LOT number/customer management number/product match confirmation, the creation of inspection reports by type, and final LOT work were completed using machines and humans to insert separators and check the quantities. There were two types of waiting for shipment: packing, which involved waiting for delivery after banding; shipment, which included delivery registration, product boarding, and the registration of delivery completion. There were also two types of shipment completion: shipment with registered customer with receiving registration completed; shipment waiting for customer completion, i.e., waiting for the customer company to complete their stocking after the shipment.

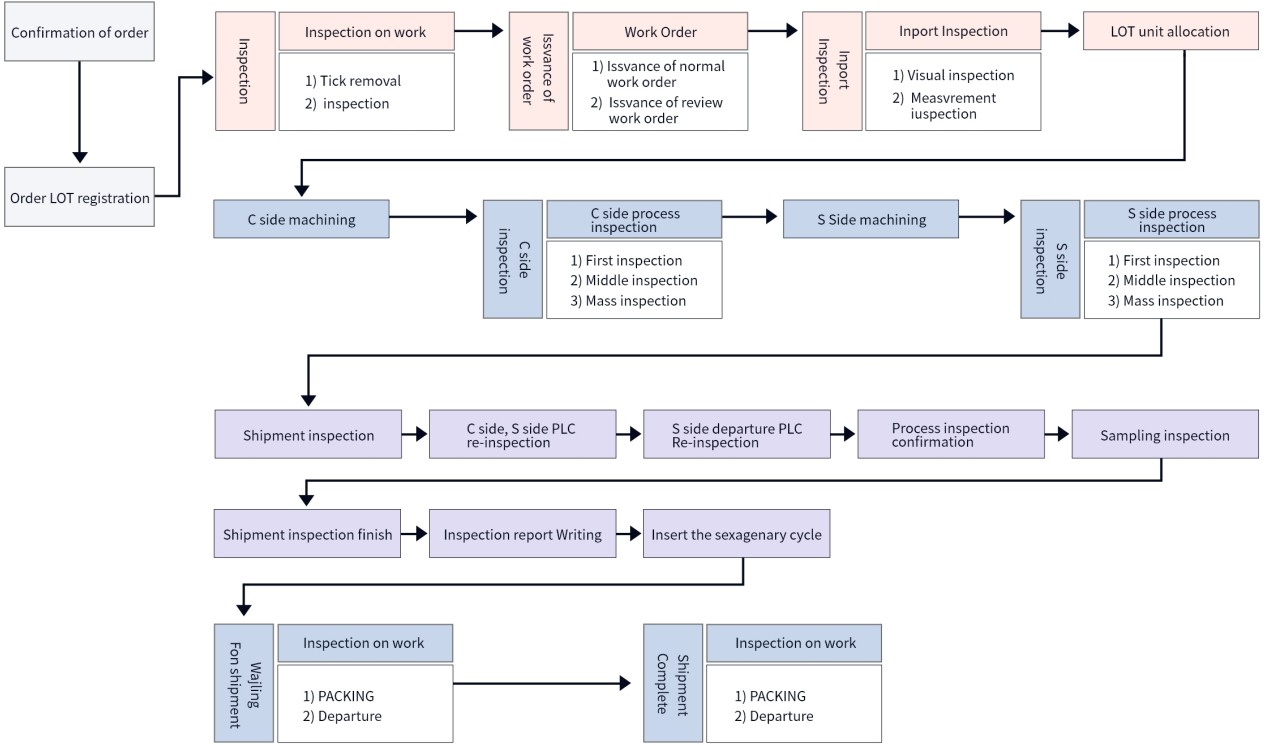

**Figure 13.** A flow chart for each major process.

## 4.4. Implementation of the NGIM HCPS 2.0-Based MES

The facility operation rate improved, as did worker awareness, by using the real-time facility allocation status analysis. The goal was to increase the facility uptime by preparing LOT and minimizing facility downtime. Figure 14 shows the status of the facility operation.

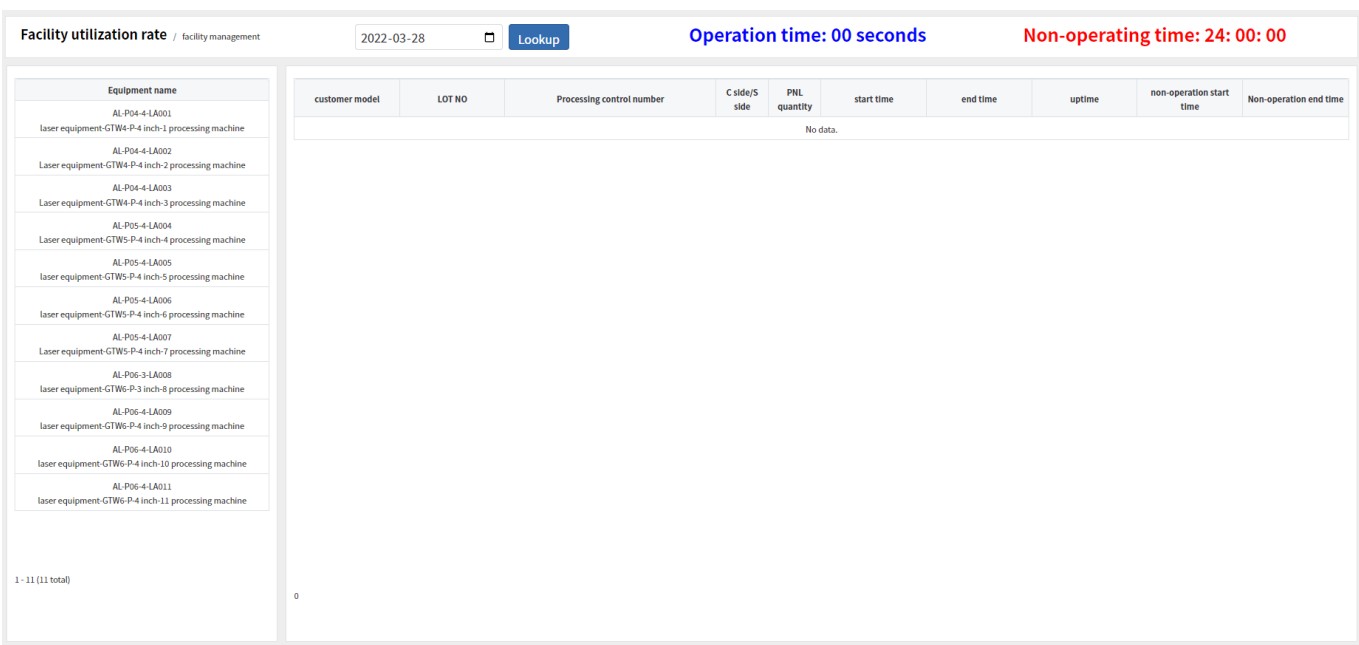

**Figure 14.** The status of the facility operation.

The qualitative target of the process defect rate was measured once a quarter and surveys and usage status interviews were conducted. This aimed to improve satisfaction by generating and improving improvement items. The quantitative performance analysis was executed once a quarter and the targets were managed by providing comparative data on the quantitative performance before and after implementation in each quarter. Figure 15 shows the status of a process inspection.

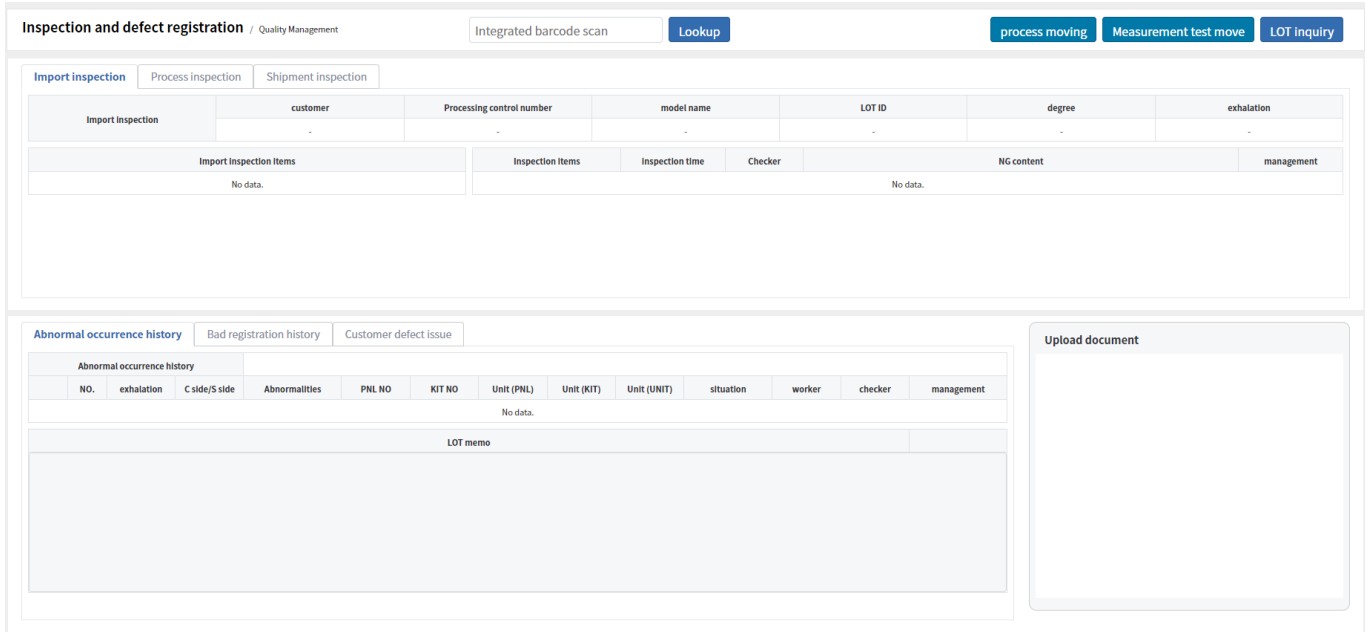

**Figure 15.** The status of a process inspection.

There was an OPC agent as a component of the IoT platform, which was installed in the processing machine. It detected changes in the log files that were targeted by the processing machine and when a change was detected, the relevant file was sent to the server. Because the machine's hardware requirements were low, it tried to use as few resources as possible. The essential requirements of the agent were Windows XP, Windows 7 or Windows 10,

Java 7 or Java 8, at least 32 MB (but recommended 128 MB) of memory, and a network environment that could communicate with the server. The agent status and transmitted log files were collected and stored on a disk by the OPC server. Figure 16 depicts the log file status for each facility, which were gathered in this manner. The OPC agent compared the collected files to those in the database and saved the differences. Tomcat 8.5, Java 8, Microsoft SQL Server 2019, and a network environment that was capable of communicating with the agent were all required for this process.

**Figure 16.** Log file status by facility.

In this study, we tried a traditional image processing method to detect holes in images. At first, when a circle was detected using the existing circle detection algorithm, the correct circle could not be detected when there was a lot of noise in the image. In addition, after applying preprocessing technology that separated the color and texture of the area in which the circle should be detected, an attempt was made to detect approximate circles using the RANSAC algorithm, but the boundary of the circles was not clear. The reasons that it was difficult to apply these traditional image processing techniques were as follows.

1. The PCB board was processed using a laser, so there were numerous heat-scorched marks around the holes; therefore, in order to detect accurate circles by excluding all of these numerous instances of noise, it was judged that the traditional algorithms for detecting images that conformed to specific ruled were not suitable;

2. The images of holes were divided into two types: TOP, in which the boundaries of the holes were clear enough to be identified with the naked eye; and BOTTOM, in which the boundaries of the holes were not easily distinguished, even by the naked eye;

3. Only the size of the hole was measured to determine whether or not it was acceptable; in the future, the goal would be to detect various types of defects, such images that contain foreign substances, images that are tilted to one side or over-processed images.

For this reason, we wanted to measure the hole sizes using the AI method. Therefore, in this study, our microscopic image analysis that was based on deep learning vision technology for real-time quality inspection used U-Net, which is a deep learning algorithm that is effective in the image segmentation of medical images. Figure 17 shows the U-Net training process. The contour detection algorithm detected an area in the image and measured the diameter after masking in the hole image. A total of 144 data points were used: 100 training data points, 28 validation data points, and 16 test data points. Figure 17A We then proceeded with the labeling to designate the holes in the original images. Figure 17B

shows a list of data points that were displayed in this way. The images in Figure 17C show the labeling data in the first row, the epoch 25 data in the second row, and the epoch 100 data in the third row. Figure 17D shows the original images, the resulting images, and the resulting images overlaid onto the original images.

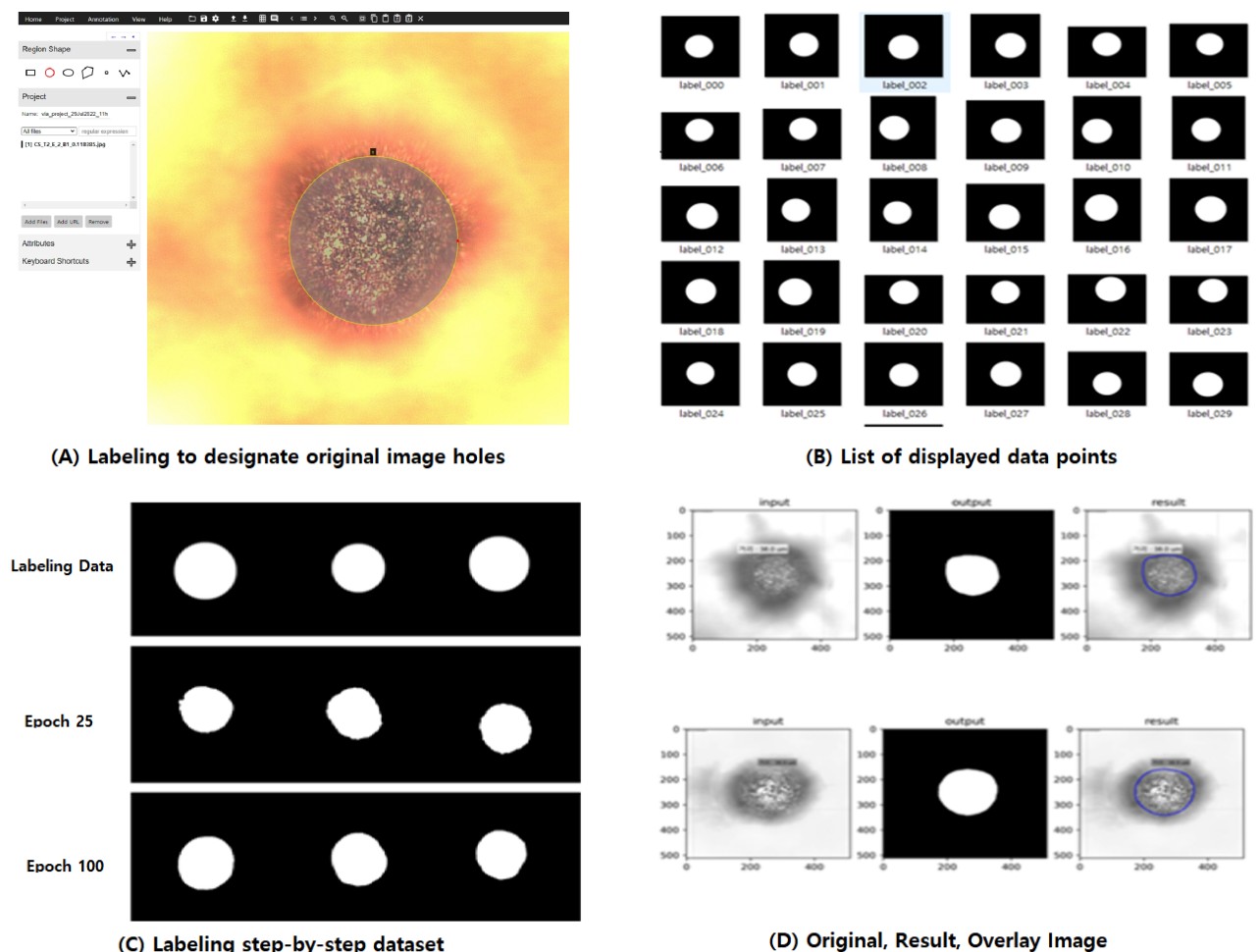

**Figure 17.** The U-Net training process.

Figure 18 shows detailed information about the hole images that were measured using the AI method. The table on the left shows the image names, LOT numbers, processing control numbers, and other detailed information about the holes. The four photos on the right show an example of an original image, the corresponding preprocessed image that measured the hole in the original image, the image that was measured by the operator, and the image and measurement values that were measured by the AI method. These images allowed the workers to monitor what the AI method was measuring.

Using data visualization, a real-time monitoring program was implemented to view the status of inspection progress by unit, the LOT status for each process, and the progress status for each LOT. Figure 19 depicts the inspection process of the HCPS unit, which calculated and displayed in real time the scheduled inspection time, inspection completion time, and remaining time for each unit inspection, including the first inspection, heavy inspection, and final inspection. This allowed the workers to keep track of whether the inspections were being completed on time.

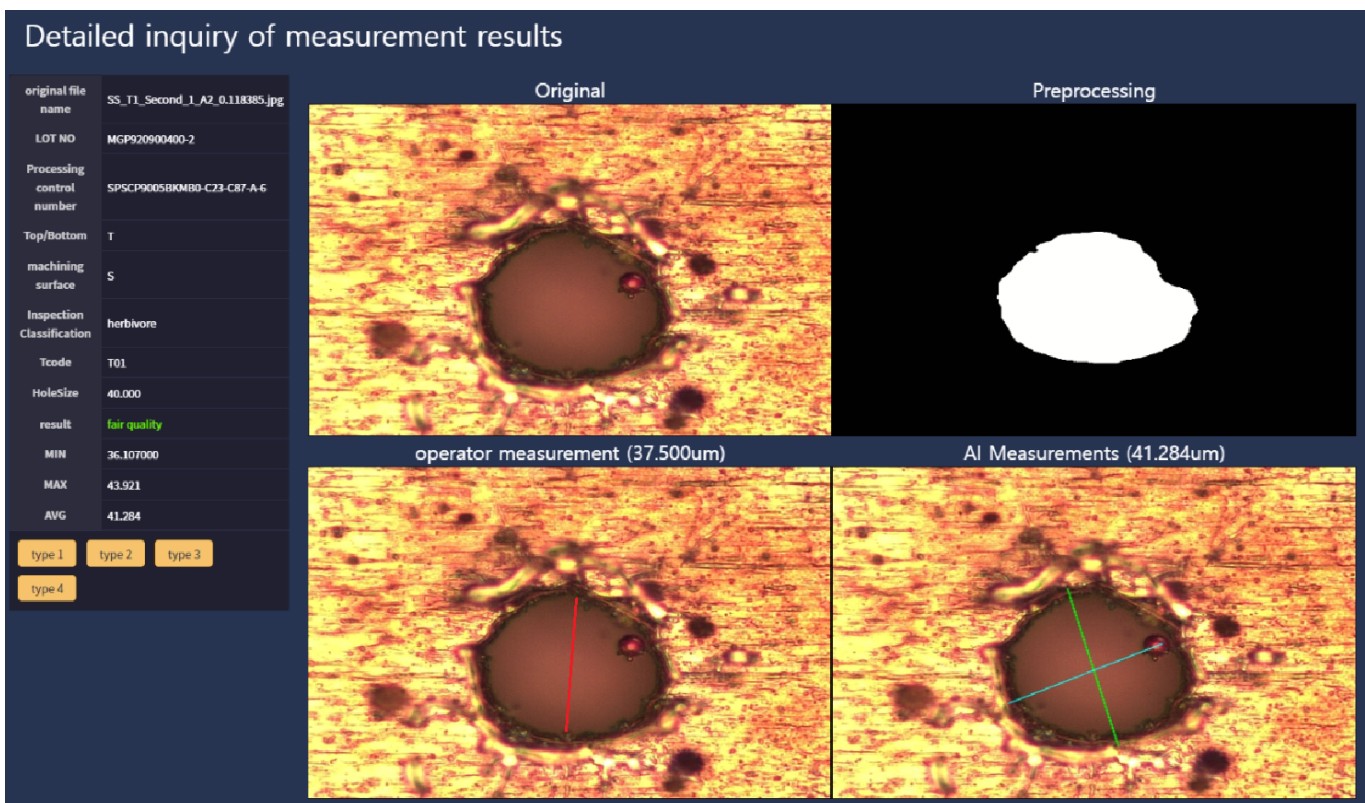

**Figure 18.** Information about an example hole that was measured using the AI method.

| May 28, 2022 at 16:38 | | | | Inspection progress screen by unit | | | | | | | | | | | | |
|---|---|---|---|---|---|---|---|---|---|---|---|---|---|---|---|---|
| | | | | split test | | | first inspection | | | heavy goods inspection | | | mass examination | | | |
| exhalation | Processing control number | LOT NO | C/S side | Scheduled time | Time remaining | completion time | Scheduled time | Time remaining | completion time | Scheduled time | Time remaining | completion time | Scheduled time | Time remaining | completion time |
| Unit 1 | 22FBO042-05-C12-C21-A-1 | 129005200C5F | SC S side | May 28th 16:12 | -25 minutes | | May 28th 16:15 | -22 minutes | | | | | | | |
| Unit 2 | 22FBO041-01-C12-C21-A-1 | 12869220010F | SC C side | May 28th at 13:34 | -183 minutes | | | | | | | | | | |
| Unit 3 | 22FBO049-02-C12-C21-A-1 | 12900560010F | SC S side | May 28th 16:13 | -24 minutes | | May 28th 16:16 | -21 minutes | | | | | | | |
| Unit 4 | SPSCP9005BKMA0-C34-C76-A-4 | MGP920914500-1 | CS C side | May 28th 04:22 | 0 minutes | May 28th 04:22 | May 28th 05:08 | -45 minutes | May 28th 05:53 | May 28th 08:13 | 39 minutes | May 28th 07:33 | May 28 at 12:04 | 144 minutes | May 28th 09:40 |
| Unit 5 | SPSCP9005BKMB0-C23-C87-A-6 | MGP921107300-2 | SC S side | May 28th 15:41 | 0 minutes | May 28th 15:42 | May 28th 16:27 | -10 minutes | | May 28th 23:20 | 402 minutes | | May 29th 03:10 | 632 minutes | |
| Unit 6 | SPSCP9005BKMB0-C23-C87-A-6 | MGP921407100-2 | CS C side | May 28th 06:05 | 0 minutes | May 28th 06:05 | May 28th 06:36 | 41 minutes | May 28th 05:55 | May 28 11:17 | 158 minutes | May 28th 08:39 | May 28th at 13:53 | 166 minutes | May 28th 11:07 |
| Unit 7 | SPSCP9005BKMB0-C23-C87-A-6 | MGP921402800-1 | CS C side | May 28th at 13:41 | 0 minutes | May 28th at 13:41 | May 28th 14:12 | -12 minutes | May 28th 14:25 | May 28th 18:54 | 135 minutes | | May 28th at 21:30 | 292 minutes | |
| Unit 8 | SPSCP9005BKMB0-C23W-C87W-A-5 | MGP921406900-2 | CS C side | May 28th 15:31 | 0 minutes | May 28th 15:31 | | | | | | | | | |
| Unit 10 | SPSCP9005BKMB0-C34-C76-A-4 | MGP921304300-1 | CS C side | May 28th 08:58 | 0 minutes | May 28th 08:58 | May 28th 09:38 | -33 minutes | May 28th at 10:11 | May 28 12:18 | 60 minutes | May 28 11:17 | May 28th 15:38 | -6 minutes | May 28th 15:44 |

**Figure 19.** The inspection process of the HCPS unit.

Figure 20 shows the LOT status for each process. It was possible to check which LOT was in each progress by monitoring the LOT status of all of the processes, from order receipt to shipment.

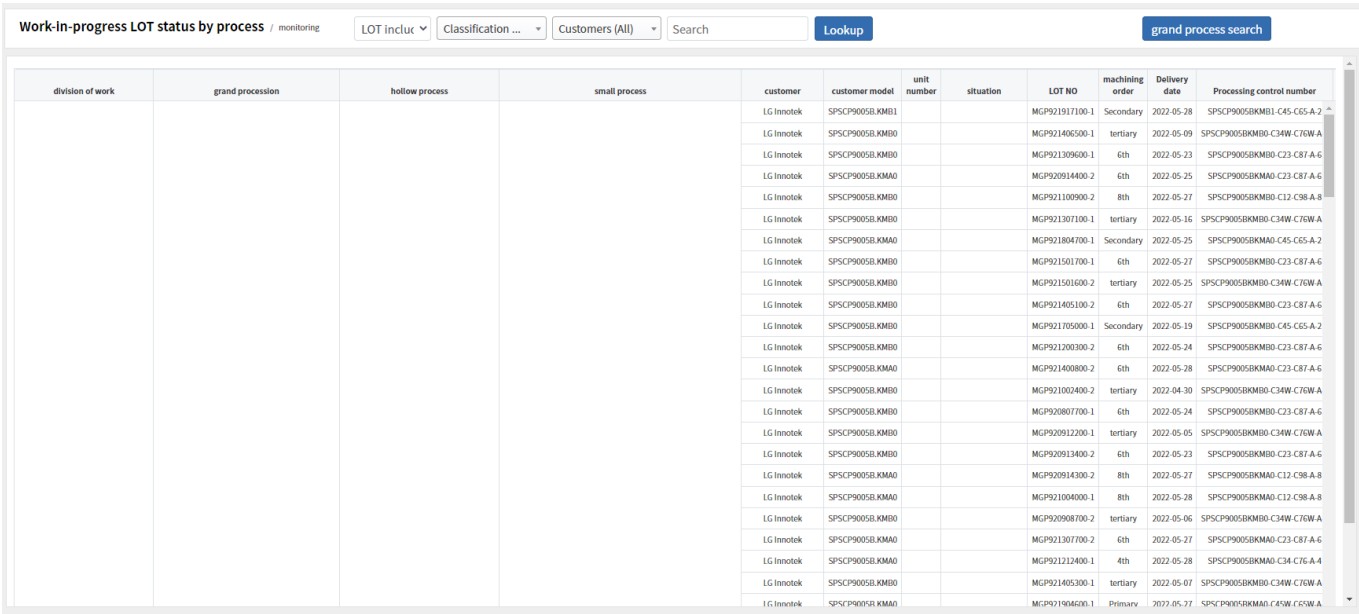

**Figure 20.** The LOT status for each process.

Figure 21 shows the progress status of each LOT. The LOTs could be searched according to date in order to monitor the status and current progress of each LOT.

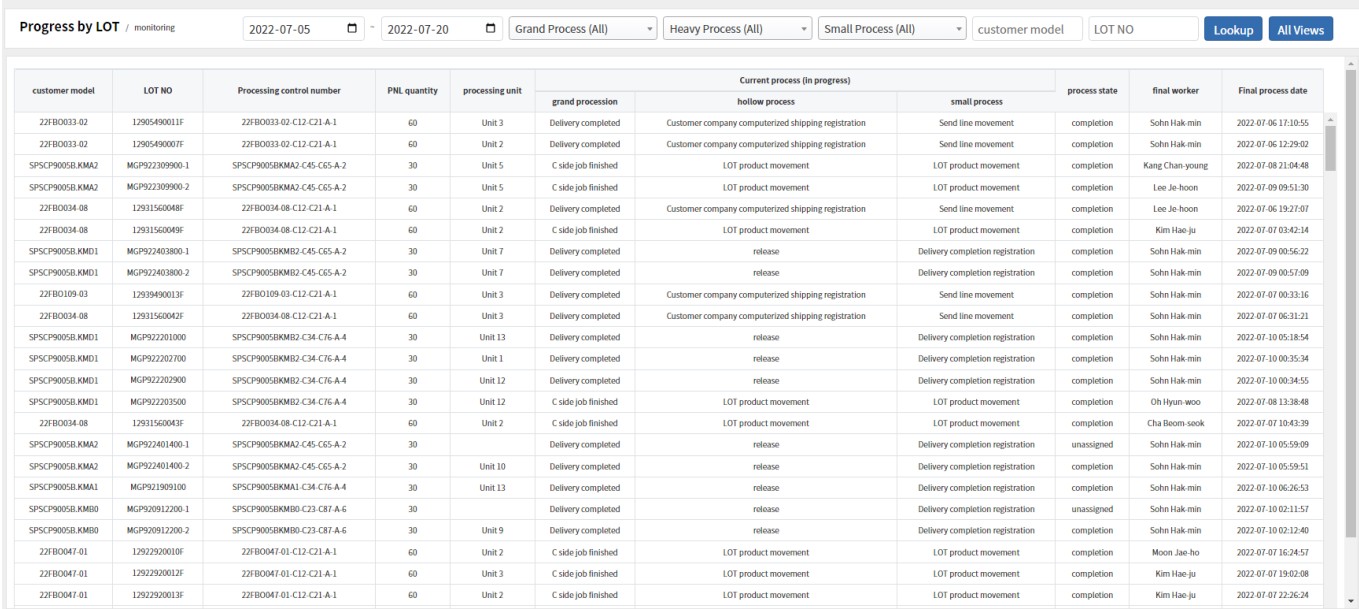

**Figure 21.** The progress status of each LOT.

Figure 22 shows the log file status for each facility and the log transmission status for each facility, which were implemented by the IoT data management. All of the data for the log file status of each facility and the log transmission status for each facility were expressed in table format so that workers could check the data by date, facility, log file, and LOT.

**LOG status management by facility** / LOG file analysis  | 2021-11-11 | - | 2021-11-14 | GTW4-P-4 inch ▾ | Unit 3 ▾ | CNDSELCT processing ▾ | LOT NO | Lookup | EXCEL Download

Search:

| LOT NO | Face processing control number | Equipment name | exhalation | DATE_TIME date and time | ROW_IDX index | C00_DATE Date | C01_TIME Time | C02_PRG part program | C03_CNDFILENAME condition file | C04_CNDNO condition number | C05_POWER output | C06_FREQ frequency | C07_PLSWID Pulse Width | C08_PLSSTAGE pulse step | C09_FEED speed | C10_PULSE number of pulses |
|---|---|---|---|---|---|---|---|---|---|---|---|---|---|---|---|---|
|  |  | GTW4 | 3 | 2021-11-11 16:30:26 | 216 | 2021-11-11 | 16:30:26 | c:\pban\prg\system\energy\energy.prg | c:\esent\CONDITION\LGIT\SPSCP90021KMA0-C23-A-1.cnd | One | 5600 | 100 | 9.0 | S | 71000 | One |
|  |  | GTW4 | 3 | 2021-11-11 16:30:26 | 217 | 2021-11-11 | 16:30:26 | c:\pban\prg\system\energy\energy.prg | c:\esent\CONDITION\LGIT\SPSCP90021KMA0-C23-A-1.cnd | 7 | 5600 | 100 | 1.0 | S | 71000 | 0 |
|  |  | GTW4 | 3 | 2021-11-11 16:25:08 | 215 | 2021-11-11 | 16:25:08 | c:\pban\prg\system\energy\energy.prg | c:\pban\prg\system\cnd\mask12-5350-COLLIMATE-ADJ.cnd | 19 | 5600 | 100 | 7.0 | S | 71000 | 2 |
|  |  | GTW4 | 3 | 2021-11-11 16:22:44 | 214 | 2021-11-11 | 16:22:44 | c:\pban\prg\system\galv_hc\galv_hc.prg | c:\pban\prg\system\cnd\mask12-5350-COLLIMATE-ADJ.cnd | One | 5600 | 100 | 100.0 | M | 71000 | 2 |
|  |  | GTW4 | 3 | 2021-11-11 16:22:40 | 213 | 2021-11-11 | 16:22:40 | c:\pban\prg\system\galv_hc\galv_hc.prg | c:\pban\prg\system\cnd\mask12-5350-COLLIMATE-ADJ.cnd | 9 | 5600 | 100 | 100.0 | M | 71000 | 2 |
|  |  | GTW4 | 3 | 2021-11-11 16:22:37 | 212 | 2021-11-11 | 16:22:37 | c:\pban\prg\system\galv_hc\galv_hc.prg | c:\pban\prg\system\cnd\mask12-5350-COLLIMATE-ADJ.cnd | 7 | 5600 | 100 | 100.0 | M | 71000 | 2 |
|  |  | GTW4 | 3 | 2021-11-11 16:22:34 | 211 | 2021-11-11 | 16:22:34 | c:\pban\prg\system\galv_hc\galv_hc.prg | c:\pban\prg\system\cnd\mask12-5350-COLLIMATE-ADJ.cnd | 6 | 5600 | 100 | 100.0 | M | 71000 | 2 |
|  |  | GTW4 | 3 | 2021-11-11 16:22:31 | 210 | 2021-11-11 | 16:22:31 | c:\pban\prg\system\galv_hc\galv_hc.prg | c:\pban\prg\system\cnd\mask12-5350-COLLIMATE-ADJ.cnd | 2 | 5600 | 100 | 100.0 | M | 71000 | 2 |
|  |  | GTW4 | 3 | 2021-11-11 16:22:29 | 209 | 2021-11-11 | 16:22:29 | c:\pban\prg\system\galv_hc\galv_hc.prg | c:\pban\prg\system\cnd\mask12-5350-COLLIMATE-ADJ.cnd | 9 | 5600 | 100 | 100.0 | M | 71000 | 2 |
|  |  | GTW4 | 3 | 2021-11-11 16:22:28 | 208 | 2021-11-11 | 16:22:28 | c:\pban\prg\system\galv_hc\galv_hc.prg | c:\pban\prg\system\cnd\mask12-5350-COLLIMATE-ADJ.cnd | 7 | 5600 | 100 | 100.0 | M | 71000 | 2 |
|  |  | GTW4 | 3 | 2021-11-11 16:22:27 | 206 | 2021-11-11 | 16:22:27 | c:\pban\prg\system\galv_hc\galv_hc.prg | c:\pban\prg\system\cnd\mask12-5350-COLLIMATE-ADJ.cnd | 2 | 5600 | 100 | 100.0 | M | 71000 | 2 |

before **One** 2 3 4 5 ⋯ 11 next

**Figure 22.** The log file status for each facility.

Figure 23 shows the operation/non-operation status and abnormal occurrence management by facility. Both operation/non-operation status and abnormal occurrence management were included so that the operation and non-operation status of all facilities could be checked at the same time to deal with abnormal occurrence management when the causes of non-operation were abnormal.

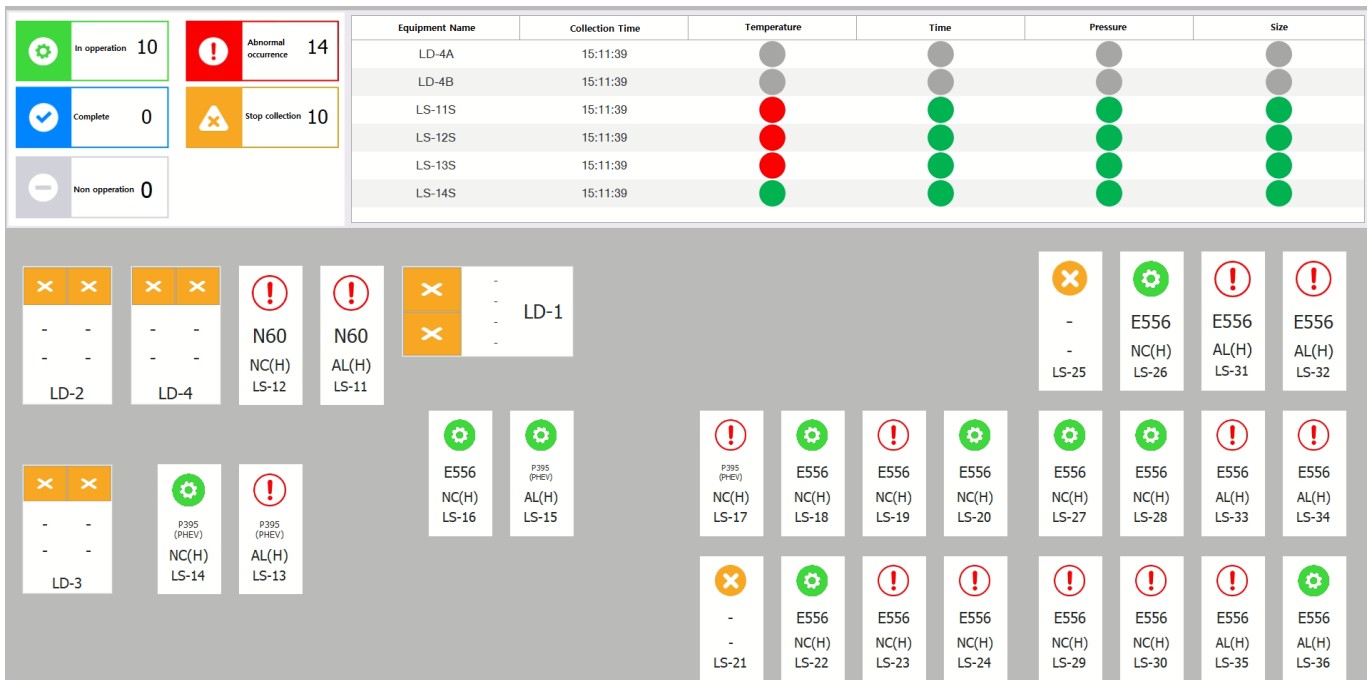

**Figure 23.** The operation/non-operation and abnormal occurrence management by facility.

Figure 24 shows the task assignment status and link to work orders by facility. The task assignment status and link to work order for each facility acted as a real-time decision support system to assist in determining which facility to assign to by checking which tasks were assigned to the current stand-by or production facilities.

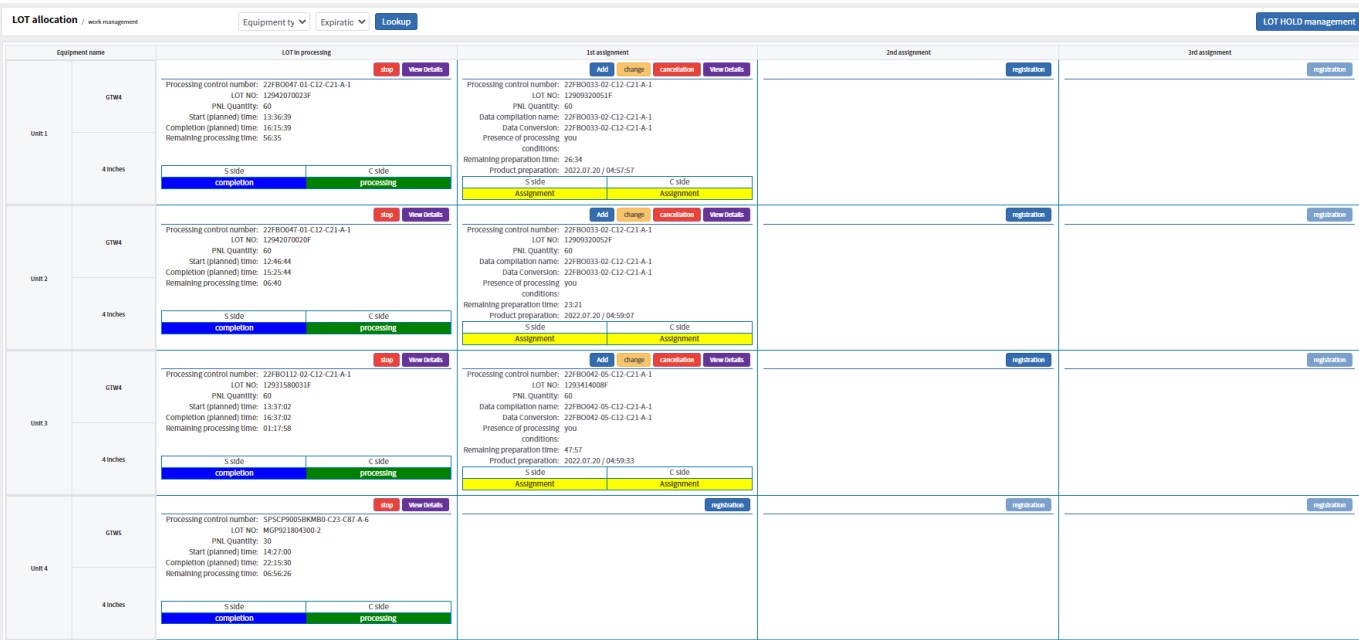

**Figure 24.** Task assignment status and link to work order by facility.

The proposed service implemented the MES system of the company, which was based on the laser process and collected real-time production results via the facility interface. The management of the real-time manufacturing conditions used the data for the facility PLC linkage, operation time/production performance/manufacturing conditions (time and processing cycles, etc.) for each facility and database, log file analysis, and statistical work. It aimed to improve corporate competitiveness by innovating the processes and quality control using statistical data. To that end, we developed an MES system that could reduce manufacturing lead times, process defect rates, and delivery times, among other KPIs, through the facility operation processes, log file collection and analysis, and data analysis for each facility.

### 4.5. Evaluation Results

Figure 25 shows the structure of the facility log file transmission and management system for data gathering. The facility log file transmission and management system was integrated into the actual implementation environment to operate three laser processing machines (GTW4, GTW5, and GTW6). Each processor had to have a DV agent installed.

The DV agents were installed in the processors. They detected changes in the log files that were targeted by each processing machine and when there was a change in a log file, that file was transmitted to the server. Because the machine's hardware requirements were low, it tried to use as few resources as possible. The server collected and saved the log files that were sent by the agent. It compared the collected files to those in the database and saved the differences. The database structure was identical to that shown in Table 2, which shows the information about the DV agents and 61 log files.

Using this, it was possible to extract the processing data from the log files of 24 laser processing machines, as shown in Table 3.

The inspection accuracy of the AI method was measured by comparing it to the accuracy of human visual inspection:

$$Accuracy = (TP + TN)/(TP + TN + FP + FN) \tag{1}$$

where *TP* is the number of true positives, *FP* is the number of false positives, *FN* is the number of false negatives, and *TN* is the number of true negatives. Figure 26 shows a diagram that contrasts the accuracy of human visual inspection and the accuracy of the AI results. In order to compare the results from the two methods, 1000 data points were

randomly selected and the accuracy of the methods was measured. The accuracy improved to 96.7% when the AI method was applied compared to 87% when human visual inspection was used.

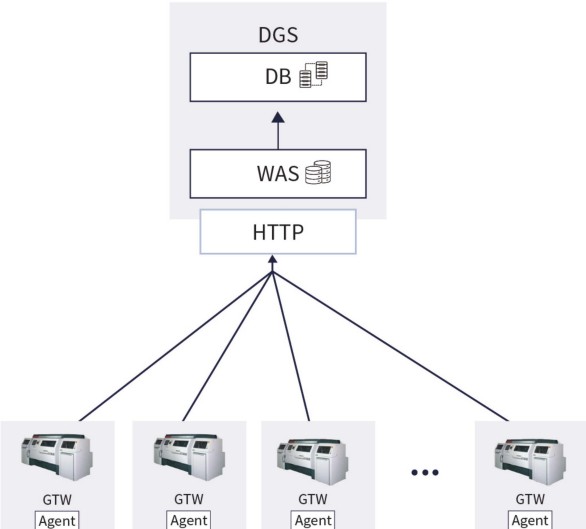

**Figure 25.** The structure of the facility log file transmission and management system.

**Table 2.** The structure of the database.

| DV Agent Information | | |
|---|---|---|
| **File** | **Data Type** | **Explanation** |
| DEVICE_NAME | VARCHAR(10) | Device name |
| DEVICE_TYPE | VARCHAR(10) | Device type |
| DEVICE_IP_ARR | VARCHAR(10) | Device IP list |
| DEVICE_DT | DATETIME2 | Heartbeat date |
| MOD_DT | DATETIME2 | Update date |
| REG_DT | DATETIME2 | Registration date |

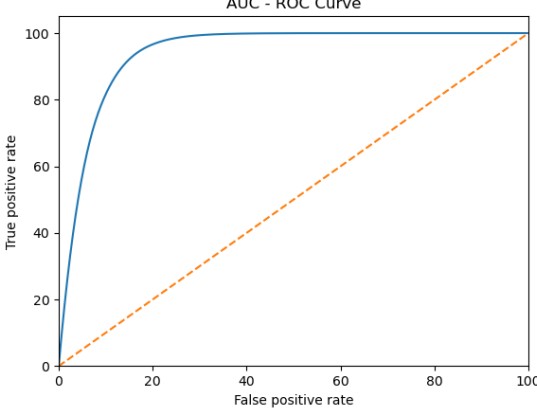

**Figure 26.** The ROC and AUC values for the results from the two methods (AI and human visual inspection).

**Table 3.** The data from the log files of 24 laser processing machines.

| File Path | GTW4 | GTW5 | GTW6 |
|---|---|---|---|
| C:/pban/LOG/aaa.log | O | O | O |
| C:/pban/LOG/aaaGosa.log | O | O | O |
| C:/pban/LOG/Almlog.log | O | O | O |
| C:/pban/LOG/cndselct.log | O | O | O |
| C:/pban/LOG/length.log | O | O | O |
| C:/pban/LOG/m370data.log | O | O | O |
| C:/pban/LOG/M377data.log | O | O | O |
| C:/pban/LOG/mainte1.log | O | O | O |
| C:/pban/LOG/maskGosa.log | O | O | O |
| C:/pban/LOG/PcDirect.log | O | O | O |
| C:/pban/LOG/pwrdata.log | O | O | O |
| C:/pban/LOG/sensor.log | O | O | O |
| C:/pban/LOG/Sinstatis.log | O | O | O |
| C:/pban/LOG/sinsyuku.log | O | O | O |
| C:/pban/LOG/table.log | O | O | O |
| C:/pban/LOG/Zaxis.log | O | O | O |
| C:/pban/LOG/Sensor/SensorHistory.log | O | O | O |
| C:/pban/LOG/ProcState.log | O | O | O |
| C:/pban/LOG/EnergyMeasure.log | O | O | O |
| C:/pban/LOG/GalvMaskMeasure.log | O | O | O |
| C:/pban/LOG/PulseEnergy.log | O | O | O |
| C:/pban/LOG/GlvCndSend.log | O | O | O |
| C:/pban/LOG/TwinScaleDsp.log | O | O | O |
| C:/pban/LOG/Marking.log | O | O | O |

Table 4 shows KPI evaluation results, in which the facility operation rate refers to the increase/decrease rate (%) of the operation time of the production equipment compared to 2021 (based on there being 24 h in a day and excluding downtime due to reduced order volumes) and the process defect rate refers to the increase/decrease in the process defect rate compared to 2021 (PPM, laser drill processing standards). For the overall achievement rate, the overall improvement rate was calculated as 26.7 × 0.7 + 40.6 × 0.3% = 30.87% and the overall goal achievement rate was calculated as 155.8 × 0.7 + 122 × 0.3% = 145.66%. In the improvement rate calculation method, the increase characteristics were calculated using (performance − present)/current × 100 (%) and the decrease characteristics were calculated using (current − performance)/current × 100 (%). In the goal achievement rate calculation method, the increase characteristics were calculated using (performance − current)/(target − present) × 100 (%) and the decrease characteristics were calculated using (current − performance)/(current − goal) × 100 (%). In the improvement/goal achievement average rate calculation method, the improvement rate was calculated using (index 1 improvement rate × weight) + (index 2 improvement rate × weight) + (index 3 improvement rate × weight). The achievement rate was calculated using (index 1 achievement rate × weight) + (index 2 achievement weight) + (index 3 achievement rate × weight).

**Table 4.** The KPI evaluation results.

| Number | Field | Key Performance Indicator | Unit | Current | Target | Performance | Improvement Rate (%) | Goal Achievement Rate (%) | Weight |
|---|---|---|---|---|---|---|---|---|---|
| 1 | P | Facility Rate (Increase Characteristics) | % | 70 | 82 | 88.7 | 26.7 | 115.8 | 0.7 |
| 2 | Q | Process Defect Rate | PPM | 300 | 200 | 178 | 40.6 | 122 | 0.3 |
| | | Total Achievement | | | | | 33.65 | 138.9 | 1 |

The benefits for production were gained from the effects of increasing worker awareness using real-time facility allocation status identification. Minimizing LOT preparation and the non-operation time of equipment by identifying equipment in a non-operational state and LOT non-assignment status improved facility operation rates and shortened facility operation times. Using systematic defect history management, process inspections could be performed by the system to identify the causes of defects, improve worker awareness, prevent unnecessary losses, and reduce defects by establishing accurate work plans and instructions and by accumulating data for each defect type. The reductions in the number of defects were identical to those that were found in the analysis.

**5. Conclusions**

NGIM, the core technology of the fourth industrial revolution, is an unprecedented technology that can be used for research tasks and overcoming challenges. NGIM faces three major challenges: system modeling, knowledge engineering, and human–machine collaboration. Smart factory systems that are designed for the PCB manufacturing industry must be consistent with the integrated control and optimization management theory, methods, and technology of processing services within the context of NGIM. To this end, a smart factory model that was based on an NGIM HCPS system that was suitable for the PCB manufacturing industry was developed. We redeployed worker roles within production site management and proposed a management method for product design, production, resource management, and smart factory utilization. A feedback mechanism was built at each stage by integrating and applying the collected data and knowledge. Finally, a "human-centric" production and management technology architecture was proposed to guide the coordination and optimization of the entire product life cycle, including design, manufacturing, operation, and maintenance. This work could provide guidelines for effective collaboration and value sharing, resource scheduling, knowledge services optimization, and guidance management methods that could adapt to dynamic market demands. The relationship between humans and cyber systems has qualitatively changed. Because of our analytical and decision-making abilities, humans can participate in the operation of cyber systems. Workers also have cognitive and learning capabilities, which can be taught to cyber systems. The primary role of machines in physical systems is to reduce specific human and cyber system resources. It is expected that this work could help to guide technical methods in the three aspects of product design and production management, resource management, and knowledge discovery and management and realize the cooperation and optimization of entire product life cycles. Smart factory systems are essentially real-time loop feedback systems. The operation and maintenance of human–cyber–physical systems are driven by information and the three systems are interconnected by high efficiency and quality.

In this study, AI technology was used as a means to realize manufacturing intelligence by creating a smart factory system that was dedicated for the next-generation PCB manufacturing industry using on an NGIM HCPS. However, operator intervention was unavoidable in the AI technology that was used in this paper and the operators had to provide high-quality data to obtain high-quality results. It was not suitable for systems in which operators directly adjust the magnification, focus, and brightness of microscopes. However, in future research, it could be possible to reduce operator intervention by pre-

training the inspection system using data that were collected from various situations, such as magnification, focus, brightness, etc. In the future, we plan to study the application of a model that can incorporate image processing technology to restore focus. Furthermore, automated optical inspection (AOI) equipment could be used to automatically measure and store images that are manually measured by operators in order to locate and capture specific holes on PCBs, thereby further reducing inspection times. As a result, we intend to investigate these issues in future studies to improve the use of manufacturing intelligence within the PCB manufacturing industry.

## 6. Study Strengths and Limitations

### 6.1. Strengths

The digital transformation (DT) of printed circuit board (PCB) manufacturers could benefit from the findings of this study. South Korea is the only country in which the top-down management of innovative manufacturing technology has been envisioned as a concept called "smart factories". In order to conceptualize innovative manufacturing technologies and develop empirical cases, it is expected that the documentation of the technologies that emerge as a result of this concept and the encouragement for follow-up research will be beneficial, not only for South Korea but also for other nations that attempt digital transformation within manufacturing industries. The field of artificial intelligence (AI) is incredibly broad and it is becoming a popular issue in many industries. Currently, there are few examples of AI solutions that are appropriate for Korean manufacturers in the PCB industry. Therefore, it is challenging to identify cases that could be systemized with research methodology since it is a relatively new attempt and there are extensive numbers of AI solutions to be considered. As a result, attempts to integrate AI technology from other fields into smart manufacturing technology (i.e., smart factories) may inspire future academic research to combine AI approaches from other manufacturing-related domains.

### 6.2. Limitations

There have not been any cases of integrated systems that can gather and analyze data that are generated from various sources at production sites (such as workers, processing equipment, quality inspection equipment, environmental facilities, and power devices) in real time, even though PCB manufacturing sites in Korea require more advanced smart factory systems. Due to this, building and implementing AI methods that are optimized for quality inspection, facility prediction/maintenance, and the detection of anomalies in production equipment, equipment control, and process control present numerous challenges. As a result, this study aimed to be the first step toward systematization research within this field by utilizing empirical technology that was directly applied to the field rather than using current research techniques, which is anticipated to have a favorable impact on future research.

Although U-Net, etc. were chosen for this study to intellectualize and advance thee quality testing for hole processing, future research should aim to achieve better performances and conduct in-depth research by adopting other algorithms, modifying parameters or developing new algorithms. It is hoped that more research and debates will be conducted in the future to develop architectures that can improve intelligent system structures and the use of AI techniques for anomaly management, facility prediction/maintenance, and process, as discussed in this paper. Additionally, the foundations and knowledge base of research within this industry will become further enriched as additional innovations are derived and published as a result of this study. Through this, we hope that more academic researchers will explore the use of AI technology in smart factories and that this paper will inspire more academic studies. The authors of this study will also contribute to this future research through their ongoing experiments.

*6.3. Opportunities*

Research on the application of OPC servers and UA that is utilized by processing manufacturers should be taken into consideration so that the work status and state of processing machines can be easily gathered and evaluated in real time. When developing intelligent smart factory systems with real-time feedback that is sent to production sites using AI methods, intelligent methods that facilitate facility prediction/preservation, abnormal occurrence management, and automatic task allocation should also be considered.

Additionally, the precise inspection of processing holes could be carried out using the AI method that was derived from the additional research methods that were mentioned in this text. Panel quality inspections could also be carried out using AOI (automatic optical inspection) systems. Through this, it is also conceivable that fresh approaches could be developed to enable the collaboration of AOI and AI solutions to create integrated smart quality inspection systems.

**Author Contributions:** Conceptualization, J.K. (Jinyoub Kim) and J.J.; methodology, J.K. (Jinyoub Kim) and D.S.; software, J.K. (Juhee Kim) and J.M.; validation, B.Y., H.K. and J.J.; formal analysis, J.K. (Jinyoub Kim) and J.M.; investigation, J.K. (Jinyoub Kim); resources, J.J.; data curation, J.K. (Jinyoub Kim); writing—original draft preparation, J.K. (Jinyoub Kim); writing—review and editing, J.J.; visualization, J.K. (Jinyoub Kim); supervision, J.J.; project administration, J.J.; funding acquisition, J.J. All authors have read and agreed to the published version of the manuscript.

**Funding:** This research was supported by the MSIT (Ministry of Science and ICT), Korea, under the ITRC (Information Technology Research Center) support program (IITP-2022-2018-0-01417) supervised by the IITP (Institute for Information & Communications Technology Planning & Evaluation). Also, this work was supported by the National Research Foundation of Korea (NRF) grant funded by the Korea government (MSIT) (No. 2021R1F1A1060054).

**Institutional Review Board Statement:** Not applicable.

**Informed Consent Statement:** Not applicable.

**Data Availability Statement:** Not applicable.

**Acknowledgments:** This research was supported by the SungKyunKwan University and the BK21 FOUR (Graduate School Innovation) funded by the Ministry of Education (MOE, Korea) and National Research Foundation of Korea (NRF) and the ITRC (Information Technology Research Center) support program (IITP-2022-2018-0-01417) supervised by the IITP (Institute for Information & communications Technology Planning & Evaluation).

**Conflicts of Interest:** The authors declare no conflict of interest.

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
