# Peer review of "Design and Implementation of an HCPS-Based PCB Smart Factory System for Next-Generation Intelligent Manufacturing"

_applsci, doi:10.3390/app12157645_

Round 1

Reviewer 1 Report

An interesting paper that provides a comprehensive treatment of intelligent manufacturing steps. The following might help improve the quality of the presented work:

 Figure 1: needs to be revised. The flow of information is missing.

English needs to be improved. Many sentence structures and grammatical mistakes.

Many acronyms are used, specially in figures, with no definition provided.

Typos: whole vs hole.

How AI is used in Figure 6? What was the AI role there?

Reference for figure 7? Is this author’s or taken from a reference? If the latter, then it should be properly referenced. This applied to figure 8 and 9 as well.

Some elaboration on data measurement explained in Figure 18 would help the reader to appreciate the innovation part of the work better.

Author Response

Please see attached for your review and comments.

Reviewer 2 Report

The paper describes the implementation of a factory system for the inspection of PCBs. Unfortunately, this paper has significant flaws. It is not written as a research paper but rather as a white paper or a manual for the factory personnel. 

Starting from the introduction the authors cited 24 documents in the first paragraph which are not research papers. Most of them are white papers and policy documents which do not offer any research insight into the methods used. 

Moving forward the authors attempt to list the contributions of the work presented in this paper. All three contributions sound useful and novel, however, they are not clearly presented in the remainder of the paper. 

The section of the related works utilises 17 citations that are randomly placed in the section without highlighting the current level of the state-of-the-art but just showcasing other methods. There is no clear indication of what the authors try to accomplish in this paper and how it stands in comparison with the state-of-the-art in the domain. 

Moving on to the methodology section, the work is presented with block diagrams and architecture flowcharts that do not depict any data exchange but rather present the high-level structure of these procedures. It is challenging for the readership to understand the contribution and be able to replicate the described methods. 

Especially the presentation of the main module, the AI method, is unclear. The presented procedure describes the detection of a hole in a PCB. There is no clear indication of why we need to implement an "AI method" to detect a hole that can be easily and accurately detected by classical image processing. What kind of data is used to train this AI model? How much data? Who collected them? What is the model's architecture? There is no indication that this is an intelligent module other than that it is named "AI method". 

The rest of the paper presents the rest of the procedures in a similar fashion as before where system diagrams and facility operation status images showcase the connection of some irrelevant modules. 

In the conclusion, another 10+ documents are cited which are originally written in Chinese and are not relevant to the presented work. It becomes suspicious why so many documents are cited without a clear purpose and without benefiting this paper. 

Overall, this paper needs serious revision, and descriptions of the actual algorithms used for all the modules, from the communication protocols to the presentation of the model architecture. 

Author Response

(The authors gave the same response as above.)

Reviewer 3 Report

Authors need to proof read the paper for English language and technical writing before submitting it for consideration to be published. There are number of incomplete sentences and sentences with words out of place. I started reading the paper but had to sop because of the inadequate writing level. Also, authors need to review figures for accuracy, for example Fig. 1, the flow should be from top left not to top left.

Author Response

(The authors gave the same response as above.)

Reviewer 4 Report

The article is current because the Design and Implementation of HCPS-Based PCB Smart Factory System for Next-generation Intelligent Manufacturing has an important meaning from the point of view of the competitiveness of organizations. My comments on the article are as follows:

a) One sentence is included twice in the opening chapter. The sentence on line 35 and 37 is the same sentence.

b) In the Introduction chapter, two paragraphs are identical. The paragraph from line 38 to 48 is conceptually exactly the same as the thoughts and expressions from lines 50 to 59. I suggest deleting one of the paragraphs.

c) The authors write on line 62 that "Currently, quality inspection is done manually, but it takes a significant amount of time". Quality inspection of which process or what kind of product is it done by hand?

d) Description of picture no. 2, I propose to break it down so that there are not only abbreviations in the name of the picture.

e) The authors write on line 206 that "Human-machine collaboration technology raises many uncertain and complex issues that intelligent manufacturing cannot solve solely through human or machine intelligence". In the article, it would be good to write specifically which authors think uncertain and complex issue.

f) Image no. 3 is not easy to read, it needs to be enlarged.

g) The sentence on line 221 begins with a small letter. It needs to be fixed.

h) In chapter 4.2, two paragraphs are again identical in thought. The paragraph from lines 426 to 444 is conceptually identical to the thought and expression of the paragraph from lines 445 to 462. I suggest deleting one of the paragraphs.

i) In chapter 4.4, two paragraphs are again identical in thought. The paragraph from line 564 to 575 is conceptually exactly the same as the thoughts and expressions from the paragraph from lines 577 to 588. I suggest deleting one of the paragraphs.

j) Image no. 21, 22, 23 and 24 are not legible. It is necessary to improve the quality of the images.

k) At line 662, the sentence is not complete.

Author Response

(The authors gave the same response as above.)

Round 2

Reviewer 2 Report

I agree with the authors' argument that we must document technologies necessary for manufacturing innovation. However, there are other venues where such documents can be disseminated, e.g. technical magazines and reach the right audience. 

An academic journal article that publishes results based on research is structured differently than the current manuscript. Even after the revisions, I believe this paper does not describe a system that can be evaluated and replicated by the research community. The revisions of the introductory section are welcome, the overall distribution of the citations has been improved, and the authors did a good job there. 

However, the section which describes the deep learning algorithm still needs major revision. As it is written now, it just describes how two algorithms that work as black boxes were combined into the so-called "AI method". This is inadequate, and the authors could have argued why they chose U-Net, what parameters they tuned and how U-Net performs compared with other network architectures. 

Overall, I still believe that his paper is not ready for publication and the authors should reconsider if they want to target an academic journal or a technical magazine for disseminating their work. 

Author Response

(The authors gave the same response as above.)
